# LATENT REWARD-GUIDED SEARCH FOR FASTER INFERENCE-TIME SCALING IN VIDEO DIFFUSION

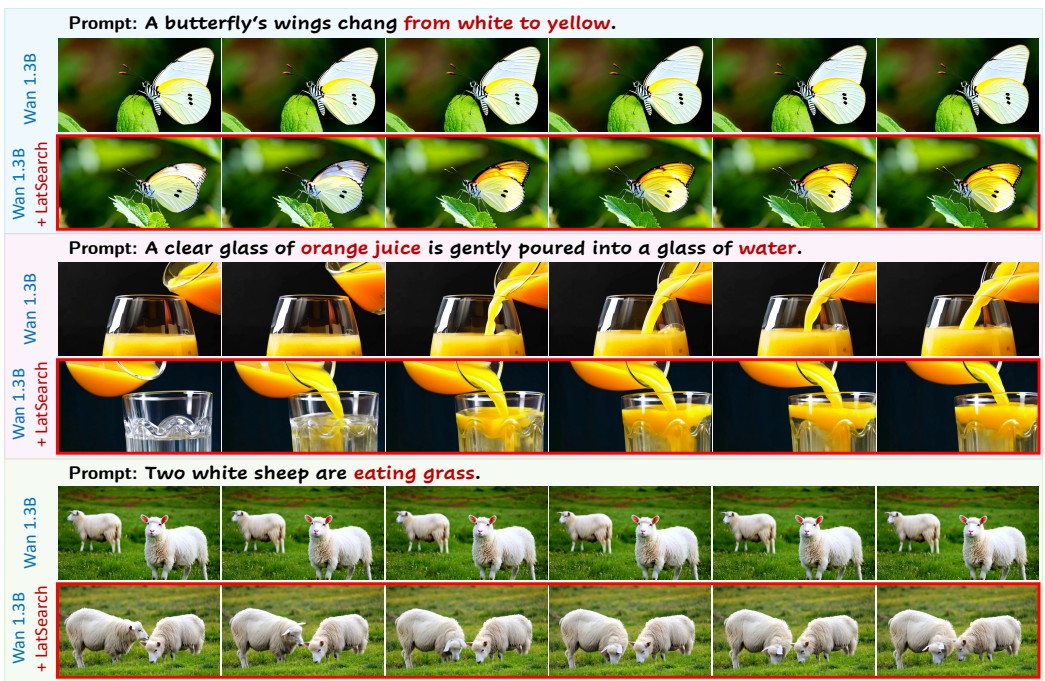

Figure 1: Text-to-video generations, comparing a vanilla model with LatSearch, a novel faster inference-time scaling method in video generation. LatSearch significantly improves sample quality by leveraging latent reward-guided computation allocation during inference, enabling early evaluation of noisy latents and the selection of credible candidates along the diffusion trajectory.

## ABSTRACT

The recent success of inference-time scaling in large language models has inspired similar explorations in video diffusion. In particular, motivated by the existence of "golden noise" that enhances video quality, prior work has attempted to improve inference by optimising or searching for better initial noise. However, these approaches have notable limitations: they either rely on priors imposed at the beginning of noise sampling or on rewards evaluated only on the denoised and decoded videos. This leads to error accumulation, delayed and sparse reward signals, and prohibitive computational cost, which prevents the use of stronger search algorithms. Crucially, stronger search algorithms are precisely what could unlock substantial gains in controllability, sample efficiency and generation quality for video diffusion, provided their computational cost can be reduced. To fill in this gap, we enable efficient inference-time scaling for video diffusion through latent reward guidance, which provides intermediate, informative and efficient feedback along the denoising trajectory. We introduce a latent reward model that scores partially denoised latents at arbitrary timesteps with respect to visual quality, motion quality, and text alignment. Building on this model, we propose LATSEARCH, a novel inference-time search mechanism that performs Reward-Guided Resampling and Pruning (RGRP). In the resampling stage, candidates are sampled according to reward-normalised probabilities to reduce over-reliance on the reward

model. In the pruning stage, applied at the final scheduled step, only the candidate with the highest cumulative reward is retained, improving both quality and efficiency. We evaluate LATSEARCH on the VBench-2.0 benchmark and demonstrate that it consistently improves video generation across multiple evaluation dimensions compared to the baseline Wan2.1 model. Compared with the state-of-the-art, our approach achieves comparable or better quality while reducing runtime by up to 79%. *The code and pre-trained reward models will be publicly available upon paper acceptance, and the core implementation is included in the supp. material.*

# 1 INTRODUCTION

Given the wide range of applications of video generation, such as video editing (Kara et al., 2024), customisation (Karras et al., 2023), image animation (Dalal et al., 2025), and world modelling (Agarwal et al., 2025), there has been a growing interest in transferring the success of inference-time scaling observed in large language models (LLMs) to diffusion-based video generation models (Liu et al., 2025a; Ma et al., 2025a). A natural way for better fidelity is to increase the number of denoising steps, which directly improves sample fidelity. However, recent research has shown that inference-time scaling extends further than simply increasing denoising steps (Ma et al., 2025a). Several studies have demonstrated the effectiveness of so-called "golden noise"—specific initial noise realisations that reliably lead to higher-quality generations—highlighting the significant role of noise initialisation in the final generation quality. (Zhou et al., 2024; Qi et al., 2024; Ban et al., 2025; Kim & Kim, 2025). Consequently, many works now attempt to allocate additional computation at inference time to improve the fidelity and consistency of generated videos (Oshima et al., 2025; He et al., 2025; Yang et al., 2025; Ma et al., 2025a).

A key problem in inference-time scaling for video diffusion lies in the inability to evaluate intermediate latents reliably. Lacking such evaluations prohibits a model from supporting more flexible strategies, such as early stopping for efficiency, resulting in errors introduced at the initial stage being accumulated throughout the long denoising trajectory. Existing methods, however, largely overlook this problem. Noise optimisation techniques bias the initialisation by injecting noise into a reference video, applying temporal warping or optical flow, or fusing frequency components of denoised latents (Chang et al., 2024; Wu et al., 2024; Burgert et al., 2025; Yuan et al., 2025; Zhang et al., 2025). However, once the trajectory begins, they lack mechanisms to monitor and correct intermediate states. Noise search methods instead generate multiple candidates and select the best based on fully decoded videos. Such models leverage strategies such as Best-of-N sampling, beam search, evolutionary algorithms, or path search (Singhal et al., 2025; Oshima et al., 2025; Yang et al., 2025; He et al., 2025; Ma et al., 2025a). These methods typically depend on reward models or verifiers, which are chosen from standard metrics (FID, IS, DINO, CLIP), or video-specific reward functions (Liu et al., 2025b). More recently, uncertainty measures derived from attention maps have also been considered (Kim & Kim, 2025). However, current noise search methods operate only on final outputs, causing them to incur high computational cost from full video decoding and suffer from reward delay (Liao et al., 2025), limiting their usefulness in guiding generation.

To address this limitation, we propose LATSEARCH, a faster and better inference-time scaling method that integrates a latent reward model into video diffusion. Unlike conventional verifiers that operate only on final decoded videos, our reward model evaluates partially denoised latents at arbitrary timesteps, providing intermediate feedback on generation progress to facilitate efficient inference-time search. By introducing process-level supervision, LATSEARCH can identify and prune low-quality candidates early, thereby reducing unnecessary denoising steps. This not only mitigates error accumulation and reward delay but also avoids the heavy cost of repeatedly decoding full videos, making a more efficient and effective search procedure possible. To enable inference-time search guided by intermediate latents, we propose a latent reward model that evaluates partially denoised latents at arbitrary timesteps with respect to visual quality, motion, and text alignment. This model directly guides the search process by scoring intermediate latents, enabling more fine-grained candidate selection than final-video evaluation. For training, we construct a dataset of (prompt, latent, timestep, video score, latent similarity) tuples, where latent similarity measures the correspondence between an intermediate latent and the final clean latent. The model is optimised with both a regression loss and a latent preference loss to improve accuracy and robustness in intermediate reward estimation. Building on this model, LATSEARCH introduces Reward-Guided Resampling and

Pruning (RGRP) to refine noise candidates during generation. The resampling stage is inspired by importance sampling: instead of deterministically picking the highest-reward candidate, we sample candidates according to reward-normalised probabilities. This prevents over-reliance on the reward model, which may be imperfect. The pruning stage, applied at the final scheduled timestep, selects the single candidate with the highest cumulative reward across timesteps, reducing redundancy and significantly lowering the computational cost of the remaining denoising process.

In summary, the main contributions of this work are as follows:

- We propose a reward model that evaluates partially denoised latents at arbitrary timesteps, providing process-level supervision on visual quality, motion quality, and text alignment. Since intermediate latents lack explicit semantics, we introduce a similarity-based grounding strategy and combine regression with preference losses to make latent-level reward estimation feasible.

- Building on this reward model, we design an inference-time scaling algorithm that incorporates intermediate supervision directly into the denoising trajectory. Candidates are probabilistically resampled according to reward-normalised weights, duplicates are removed via uniqueness pruning, and cumulative rewards are used for final selection, making latent reward actionable for efficient search.

- On VBench2.0, LATSEARCH consistently improves video generation across creativity, commonsense, controllability, human fidelity, and physics. Compared with state-of-the-art inference-time scaling methods, our approach achieves comparable or better quality while reducing runtime by up to 79%.

## 2 RELATED WORK

We review related work in two areas most relevant to our study: video generation methods and inference-time scaling for diffusion models.

**Video Generation.** Early studies on video generation explored diverse paradigms, including VAEs (Bhagat et al., 2020; Skorokhodov et al., 2022), GANs (Hsieh et al., 2018; Brooks et al., 2022), and autoregressive models (Deng et al., 2024; Gu et al., 2025). More recently, diffusion-based approaches have become the dominant paradigm (Guo et al., 2024; Yang et al., 2024; Dalal et al., 2025), achieving superior visual fidelity and scalability by extending the success of image diffusion models. Progress in video diffusion has generally followed two directions: (i) foundational models, which adapt image diffusion architectures to the temporal domain, and (ii) enhancements, which improve fidelity, efficiency, and controllability. In the text-to-video (T2V) setting, early models extended U-Net–based image diffusion backbones with temporal modules, as in Stable Video Diffusion (SVD) (Blattmann et al., 2023). Subsequent works introduced improved training strategies for temporal coherence and motion quality, exemplified by ModelScope (Wang et al., 2023), VideoCrafter (Chen et al., 2023a), and AnimateDiff (Guo et al., 2024). More recently, Diffusion Transformers (DiTs) (Peebles & Xie, 2023) have emerged as stronger backbones, offering improved scalability and modelling capacity. At the same time, training objectives have evolved beyond conventional denoising losses toward flow-based formulations, enabling more efficient and stable optimisation. Works such as UViT (Bao et al., 2023) and Gentron (Chen et al., 2024) first demonstrated the feasibility of Transformer-only backbones, inspiring large-scale systems like Hunyuan Video (Kong et al., 2024), CogVideoX (Yang et al., 2024), and Wan (Wan et al., 2025). Building on these foundation models, subsequent research has explored generating longer videos (Qiu et al., 2023; Ma et al., 2025b; Ouyang et al., 2025), improving temporal coherence (Luo et al., 2025; Peruzzo et al., 2025; Nam et al., 2025), leveraging human feedback (Yuan et al., 2024; Hiranaka et al., 2024; Liu et al., 2025b; Zhu et al., 2025), enhancing controllability (Chen et al., 2023b; Xiao et al., 2024), and improving generation efficiency (Sun et al., 2025). In our work, we build on the state-of-the-art Wan model as the foundation for video generation, and investigate inference-time scaling through latent reward guidance. While prior work has explored latent reward models for few-shot video generation (Ding et al., 2024), they are limited to evaluating the final denoised latent. In contrast, our approach evaluates intermediate states across denoising timesteps, enabling more fine-grained and effective guidance.

**Inference-Time Scaling for Diffusion Models.** There is growing interest in inference-time scaling for diffusion models, inspired by its success in large language models (LLMs) (Liu et al., 2025a;

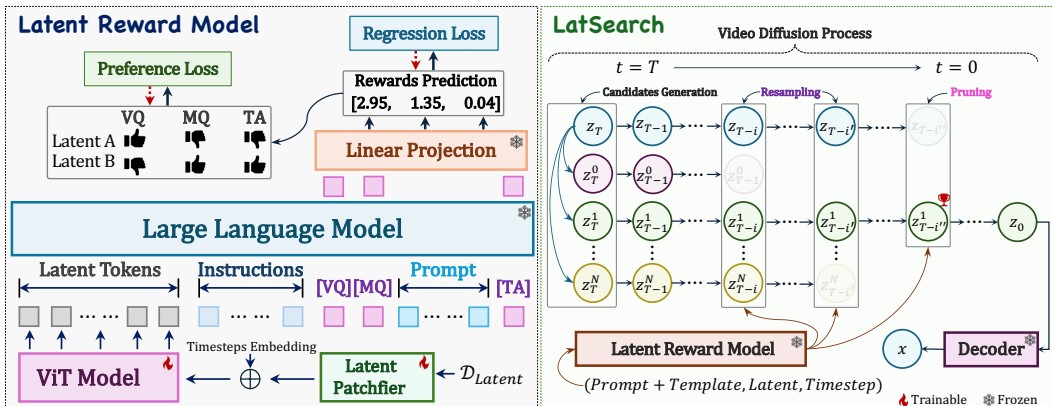

Figure 2: An overview of a latent reward model (left) and the proposed latent reward-guided inference-time search method, LATSEARCH (right). On the left, input latent tokens are patchified, fused with timestep embeddings, and projected by a ViT encoder. Together with instruction tokens, text prompts, and special query tokens ([VQ], [MQ], [TA]), these form the input to a large language model. The model is trained using a combination of regression and preference losses. On the right, LATSEARCH maintains multiple candidate trajectories during a diffusion process. Candidates are periodically scored by the latent reward model, resampled with uniqueness to encourage diversity, and finally pruned based on cumulative rewards before decoding into the final video.

Ma et al., 2025a). A straightforward scaling strategy is to increase the number of denoising steps, which generally improves sample fidelity. However, recent studies show that inference-time scaling extends far beyond this (Ma et al., 2025a). In particular, the choice of initial noise has been identified as a key factor in generation quality, with the notion of "golden noise" underscoring its importance (Zhou et al., 2024; Qi et al., 2024; Ban et al., 2025; Kim & Kim, 2025). As a result, many methods now dedicate additional computation at inference time to either optimise the initial noise or search for better noise samples (Oshima et al., 2025; He et al., 2025; Yang et al., 2025; Ma et al., 2025a). Noise optimisation approaches inject priors into noise initialisation, for example by adding noise to a reference video (Zhang et al., 2025), warping noise via temporal correlation or optical flow (Chang et al., 2024; Burgert et al., 2025), or fusing frequency components of denoised latents (Wu et al., 2024; Yuan et al., 2025). While fusion-based methods avoid reliance on external inputs, they often require more iterations. Noise search approaches instead generate multiple candidates and evaluate the resulting videos. Techniques include Best-of-N sampling (Singhal et al., 2025), beam search (Oshima et al., 2025; Yang et al., 2025), evolutionary search (He et al., 2025), and search-over-path strategies (Ma et al., 2025a). Candidate evaluation typically relies on reward models or verifiers, ranging from traditional metrics such as FID (Heusel et al., 2017), IS (Salimans et al., 2016), DINO (Caron et al., 2021), and CLIP (Radford et al., 2021), to video-specific reward models designed for temporal quality assessment (Liu et al., 2025b). Beyond explicit search, recent work has also explored estimating noise quality directly from model attention maps using uncertainty-based measures (Kim & Kim, 2025). Existing inference-time scaling methods optimise or search over initial noise and evaluate only final videos, lacking intermediate guidance. In contrast, we propose a latent reward model that assesses partially denoised latents at arbitrary timesteps, providing fine-grained feedback on visual quality, motion, and text alignment.

## 3 METHOD

Our approach, LATSEARCH, integrates a latent reward model with a reward-guided search mechanism to scale video diffusion at inference time. In this section, we first review the preliminaries of video diffusion models, then introduce a latent reward model that provides intermediate evaluations throughout the denoising process, and finally describe how these signals are incorporated into a reward-guided search strategy for efficient and high-quality video generation.

### 3.1 PRELIMINARIES

Latent text-to-video diffusion models encode an $F$-frame clean video $\{\boldsymbol{x}^{(i)}\}_{i=1}^N \in \mathbb{R}^{F \times C \times H \times W}$ into latent representations $\{\boldsymbol{z}_0^{(i)}\}_{i=1}^N$ using an encoder $\mathcal{E}$, where $C, H, W$ denote the channel, height,

and width of each frame. For convenience, we denote $\boldsymbol{z}_0 = \{\boldsymbol{z}_0^{(i)}\}_{i=1}^N \sim p_0(\boldsymbol{z})$. The forward diffusion process (Dhariwal & Nichol, 2021) gradually perturbs $\boldsymbol{z}_0$ into a noised latent $\boldsymbol{z}_t$ according to

$$q(\boldsymbol{z}_t \,|\, \boldsymbol{z}_0) = \mathcal{N}\big(\boldsymbol{z}_t; \sqrt{1-\bar{\alpha}_t}\,\boldsymbol{z}_0,\ \bar{\alpha}_t\mathbf{I}\big), \tag{1}$$

where $\bar{\alpha}_t$ is the noise schedule coefficient at timestep $t$.

In text-to-video generation, a prompt $P$ is encoded into a condition $\boldsymbol{c} = \mathcal{E}_{\text{text}}(P)$, which guides the denoising process. The training objective minimises the mean squared error between the true noise $\boldsymbol{\epsilon} \sim \mathcal{N}(\mathbf{0}, \mathbf{I})$ and the predicted noise:

$$\mathbb{E}_{\boldsymbol{z}_0, \boldsymbol{\epsilon}, t, c}\Big[\ \|\boldsymbol{\epsilon} - \boldsymbol{\epsilon}_\theta(\boldsymbol{z}, t, \boldsymbol{c})\|^2\ \Big], \tag{2}$$

where $\boldsymbol{\epsilon}_\theta(z_t, t, \boldsymbol{c})$ is the noise predictor parameterized by $\theta$.

After training, we generate videos by solving the reverse diffusion ODE (Song et al., 2020b) using the UniPC sampler (Zhao et al., 2023), a second-order predictor–corrector framework designed for fast and accurate sampling. The ODE is expressed as

$$\frac{d\boldsymbol{z}_t}{dt} = f_\theta(\boldsymbol{z}_t, t, \boldsymbol{c}), \tag{3}$$

where $f_\theta$ is derived from $\boldsymbol{\epsilon}_\theta$ under the probability-flow formulation. To strengthen text guidance, we apply classifier-free guidance (CFG) (Ho & Salimans, 2022):

$$\boldsymbol{\epsilon}_\theta^w(\boldsymbol{z}_t, t, \boldsymbol{c}) = \boldsymbol{\epsilon}_\theta(\boldsymbol{z}_t, t, \varnothing) + w\big[\boldsymbol{\epsilon}_\theta(\boldsymbol{z}_t, t, \boldsymbol{c}) - \boldsymbol{\epsilon}_\theta(\boldsymbol{z}_t, t, \varnothing)\big], \tag{4}$$

where $w \in \mathbb{R}_{\geq 0}$ is the guidance scale and $\varnothing$ denotes the null text prompt.

At each denoising step $t_s \to t_{s-1}$ with step size $h_s$, UniPC applies a second-order update of the form

$$\boldsymbol{z}_{t_{s-1}} = \boldsymbol{z}_{t_s} + \frac{h_s}{2}\Big(f_\theta(\boldsymbol{z}_{t_s}, t_s, \boldsymbol{c}) + f_\theta(\tilde{\boldsymbol{z}}_{t_{s-1}}, t_{s-1}, \boldsymbol{c})\Big), \tag{5}$$

where $\tilde{\boldsymbol{z}}_{t_{s-1}} = \boldsymbol{z}_{t_s} + h_s f_\theta(\boldsymbol{z}_{t_s}, t_s, \boldsymbol{c})$ serves as a predictor. Finally, the terminal latent $\boldsymbol{z}_0$ is decoded by $\mathcal{D}$ into an $N$-frame RGB video, $\{\boldsymbol{x}^{(i)}\}_{i=1}^N = \mathcal{D}(\boldsymbol{z}_0)$.

## 3.2 Latent Reward Model

To enable intermediate evaluations during video diffusion, we design a latent reward model that can assign quality scores to partially denoised latents without decoding to the video space. We first describe the construction of the training dataset, then detail the architecture of the reward model, and finally present the objective function used to optimise it.

**Latent Reward Data Construction.** Most reward assessors operate on rendered videos and return video-level scores, making it nontrivial to supervise rewards directly on latent representations. Concretely, given a prompt $p$ and the final denoised ("clear") latent $\boldsymbol{z}_0$, the video-level reward vector is obtained on the decoded video

$$\boldsymbol{r} = \big(r^{\text{VQ}}, r^{\text{MQ}}, r^{\text{TA}}\big)^\top = \mathcal{R}\big(\mathcal{D}(\boldsymbol{z}_0), p\big), \tag{6}$$

where $\mathcal{D}$ is the decoder, $p$ is the prompt, and $\mathcal{R}$ denotes external verifiers or human annotations for visual quality (VQ), motion quality (MQ), and text alignment (TA) (Liu et al., 2025b). However, our reward model must evaluate intermediate latents $\boldsymbol{z}_t$ extracted along the denoising trajectory. Since direct latent-level ground truth is unavailable, we propose a similarity-grounded credit assignment. Specifically, we ground video-level rewards to intermediate latents by measuring how much an intermediate latent $\boldsymbol{z}_t$ "contributes" to the final clear latent $\boldsymbol{z}_0$ via its similarity to the clear latent. We define a cosine-based similarity, rescaled to $[0, 1]$:

$$s_t = \frac{1}{2}\left(1 + \frac{\langle \boldsymbol{z}_t, \boldsymbol{z}_0 \rangle}{\|\boldsymbol{z}_t\|_2\,\|\boldsymbol{z}_0\|_2}\right) \in [0, 1]. \tag{7}$$

The similarity $s_t$ quantifies how close $\boldsymbol{z}_t$ is to $\boldsymbol{z}_0$ in a task-relevant representation. Finally, we assign latent-level targets by crediting each dimension of the video-level reward proportionally to $s_t$:

$$\tilde{\boldsymbol{r}}_t = s_t \cdot \boldsymbol{r} = \big(s_t\, r^{\text{VQ}},\ s_t\, r^{\text{MQ}},\ s_t\, r^{\text{TA}}\big)^\top. \tag{8}$$

This yields the latent reward dataset

$$\mathcal{D}_{\text{latent}} = \big\{ (\boldsymbol{z}_t, \, p, \, \tilde{\boldsymbol{r}}_t, \, t) : t \in \mathcal{T} \big\}, \tag{9}$$

where $\mathcal{T}$ is the set of sampled timesteps.

**Model Architecture.** The reward model takes three inputs: a latent representation $\boldsymbol{z}_t \in \mathbb{R}^{F/4 \times C' \times H/8 \times W/8}$ (where $F$, $C'$, $H$, and $W$ denotes video frames number, latent channel, video height, and video width), the denoising step $t$, and the text prompt $p$. These inputs are unified into a token sequence and processed by a transformer-based backbone. *Latent tokens:* The intermediate latent tensor $\boldsymbol{z}_t$ is patchified by a lightweight 3D convolutional encoder, which partitions the spatiotemporal volume into non-overlapping blocks and projects them into embedding vectors. This yields a sequence of video tokens that capture both spatial appearance and temporal motion. *Step embedding:* The denoising step $t$ is mapped to a learnable embedding vector $\mathbf{e}_t$ through an embedding layer. This embedding is concatenated with the token sequence to provide the model with explicit temporal information about the diffusion process. *Prompt tokens:* The text prompt $p$ is formatted using an instruction-style template designed for reward modeling (Liu et al., 2025b), and tokenized into instruction tokens. Special query tokens [VQ], [MQ], and [TA] are appended to the sequence to request predictions for visual quality, motion quality, and text–video alignment, respectively. The overall structure is illustrated on the left of Figure 2.

The unified token sequence is then processed by a transformer-based backbone, which outputs hidden states for the query tokens. Finally, the hidden states corresponding to the three reward query tokens are extracted and projected by a linear layer to obtain scalar scores:

$$\hat{\boldsymbol{r}} = \text{Linear}\Big( h^{[\text{VQ}]}, \, h^{[\text{MQ}]}, \, h^{[\text{TA}]} \Big), \tag{10}$$

where $h^{[\cdot]}$ denotes the hidden state of each query token. This yields the predicted rewards $(\hat{r}^{\text{VQ}}, \, \hat{r}^{\text{MQ}}, \, \hat{r}^{\text{TA}})$.

**Training Objective.** A latent reward model $R_\psi$ is optimised with a combination of a regression loss and a preference loss, as summarised in Algorithm 1.

Given the latent tensor $\boldsymbol{z}_t$, prompt $p$, and denoising step $t$, the model predicts reward scores $\hat{\boldsymbol{r}} = R_\psi(\boldsymbol{z}_t, p, t)$ across three dimensions. Since the latent reward dataset $\mathcal{D}_{\text{latent}} = \{(\boldsymbol{z}_t, p, \tilde{\boldsymbol{r}}_t, t)\}$ already incorporates the similarity weighting, the regression target for each dimension is $\tilde{r}_t^d$, where $d \in \{\text{VQ}, \text{MQ}, \text{TA}\}$. The regression loss is therefore

$$\mathcal{L}_{\text{reg}}^d = \big\| \hat{\boldsymbol{r}}^d - \tilde{\boldsymbol{r}}_t^d \big\|_2^2. \tag{11}$$

While regression provides absolute supervision, it does not enforce relative ordering among candidates. Inspired by reinforcement learning from human feedback (RLHF) (Ouyang et al., 2022), we introduce a preference loss that encourages the model to predict higher scores for better samples. For each pair $(i, j)$ in a minibatch, we define

$$\Delta \hat{\boldsymbol{r}}_{ij}^d = \hat{\boldsymbol{r}}_i^d - \hat{\boldsymbol{r}}_j^d, \qquad \boldsymbol{y}_{ij}^d = \mathbb{I}[r_i^d > r_j^d], \tag{12}$$

where $\boldsymbol{y}_{ij}^d$ denotes the ground-truth preference label for pair $(i, j)$ in dimension $d$. The preference loss is then formulated as:

$$\mathcal{L}_{\text{pref}}^d = \frac{1}{|\mathcal{P}_d|} \sum_{(i,j) \in \mathcal{P}_d} \log\big(1 + \exp\big(-(2\boldsymbol{y}_{ij}^d - 1)\,\Delta \hat{\boldsymbol{r}}_{ij}^d\big)\big), \tag{13}$$

where $\mathcal{P}_d$ is the set of preference pairs. This is equivalent to applying binary cross-entropy on pairwise score differences.

The final loss is a weighted sum over dimensions $d$ and both objectives:

$$\mathcal{L} = \sum_{d \in \{\text{VQ}, \text{MQ}, \text{TA}\}} \big(\lambda_{\text{reg}}^d \, \mathcal{L}_{\text{reg}}^d + \lambda_{\text{pref}}^d \, \mathcal{L}_{\text{pref}}^d\big). \tag{14}$$

We optimise the reward model parameters $\psi$ using stochastic gradient descent. Regression loss anchors the predictions to absolute reward magnitudes, while preference loss shapes the reward landscape to preserve relative orderings, analogous to the role of preference modelling in RLHF.

### 3.3 LATENT SEARCH WITH REWARD-GUIDED RESAMPLING AND PRUNING

We consider inference-time search as an importance-sampling-inspired procedure on latent trajectories. Standard samplers such as DDIM (Song et al., 2020a) or UniPC (Zhao et al., 2023) follow a single trajectory, which may fall into suboptimal modes. To improve robustness, we maintain a set of $N$ candidate trajectories, inspired by sequential Monte Carlo (SMC) methods (Doucet et al., 2001), and iteratively refine this set during denoising. The algorithm of the proposed latent search method is depicted in Appendix Algorithm 2.

**Candidate generation.** At the initial timestep $T$, we generate candidates by perturbing a base Gaussian noise $z_T^{(0)}$ with isotropic perturbations $\epsilon_i \sim \mathcal{N}(\mathbf{0}, \mathbf{I})$:

$$z_T^{(i)} = \sqrt{1 - \eta^2}\, z_T^{(0)} + \eta\, \epsilon_i, \qquad i = 1, \ldots, N, \tag{15}$$

where $\eta$ controls the candidate diversity. Each candidate is evolved independently with a diffusion sampler.

**Reward-guided resampling with uniqueness.** Let $\mathcal{Z}_t = \{(z_t^{(i)}, \sigma_i)\}_{i=1}^N$ be the candidate set at step $t$, where $\sigma_i$ is the seed identity of candidate $i$ (determined by its initial noise). At scoring steps $t \in \mathcal{S}$, where $\mathcal{S}$ is the search schedule, we obtain rewards $r_i^{(t)} = R_\psi(z_t^{(i)}, p, t)$ and convert them to normalized weights

$$\pi_i^{(t)} = \frac{\exp(\tau\, r_i^{(t)})}{\sum_{k=1}^N \exp(\tau\, r_k^{(t)})}, \qquad \sum_i \pi_i^{(t)} = 1, \tag{16}$$

where $\tau > 0$ is a temperature. We then draw $\boldsymbol{n}^{(t)} \sim \mathrm{Multinomial}\left(N;\, \pi_1^{(t)}, \ldots, \pi_N^{(t)}\right)$ with replacement, and keep the unique seeds only:

$$\mathcal{I}^{(t)} = \mathrm{supp}\left(\boldsymbol{n}^{(t)}\right) = \{\, i \mid n_i^{(t)} > 0 \,\}, \qquad \mathcal{Z}_t^+ = \{\, (z_t^{(i)}, \sigma_i) : i \in \mathcal{I}^{(t)} \,\}. \tag{17}$$

The uniqueness operator, $\mathrm{supp}(\cdot)$, removes multiplicities (duplicates of the same seed) and thus avoids wasting compute on identical trajectories, in which the survival probability of candidate $i$ after uniqueness is $1 - \left(1 - \pi_i^{(t)}\right)^N$. This increases monotonically with its weight $\pi_i^{(t)}$.

Between two scoring steps, there are many intermediate denoising updates, during which candidates evolve independently via the diffusion sampler.

**Cumulative weighting and final pruning.** We accumulate evidence across scoring steps via an additive criterion $c_i^{(t)} = c_i^{(t-1)} + \pi_i^{(t)}$, in which $c_i^{(0)} = 0$. At the final scheduled step $t' = \max(\mathcal{S})$, if multiple candidates remain, we prune by selecting the seed with the highest cumulative weight:

$$i^\star = \arg\max_{i \in \mathcal{I}^{(t')}} c_i^{(t')}, \qquad z_0 = z_0^{(i^\star)}. \tag{18}$$

The surviving latent $z_0$ is then decoded into the final video.

Overall, LATSEARCH follows the spirit of importance sampling: multiple candidate trajectories are proposed in latent space, weighted according to reward scores that act as surrogate importance factors, and resampled to balance exploration and exploitation. The final pruning step selects the most consistently high-reward trajectory, yielding the decoded video.

## 4 EXPERIMENTS

In this section, we evaluate the efficacy of our LATSEARCH through extensive experiments on a large-scale text-to-video generation task. We will first detail the implementations and then compare our method to other state-of-the-art inference-time scaling methods for video generation. We finally present the ablation studies. In addition, we provide more comprehensive comparisons of text-to-video generation results between the baseline model and our LATSEARCH in the Appendix.

### 4.1 EXPERIMENTAL SETTINGS

**Implementations.** We adopt Qwen2-VL-3B (Wang et al., 2024) as the backbone for our reward model, owing to its strong performance on multimodal understanding tasks and its suitability for video–language alignment. To construct the latent reward pairs dataset, we first sample 1,000 text

Table 1: Comparison of inference-time scaling methods for video generation on VBench-2.0. The table includes both optimisation-based and search-based approaches. $^\dagger$ indicates the use of DPM-Solver++ (Lu et al., 2025). **Bold** numbers indicate the best results, while underlined numbers indicate the second-best. Red text denotes performance degradation or more than $3\times$ additional compute, whereas Blue text denotes performance improvement with less than $3\times$ additional compute.

| Methods | Creativity | Commonsense | Controllability | Human Fidelity | Physics | Average | Inference Time (s) |
|---|---|---|---|---|---|---|---|
| Baseline | 53.81 | 55.63 | 21.99 | 82.11 | 45.98 | 51.90 | $77.21 \pm 0.26$ |
| FreeInit [ECCV'24] | 46.80 | 59.41 | 22.03 | 85.22 | 35.65 | $49.82_{(-2.08)}$ | $308.87 \pm 0.22_{(\times 4.00)}$ |
| FreqPrior [ICLR'25] | 53.70 | 55.61 | 20.35 | 83.61 | 38.60 | $50.37_{(-1.53)}$ | $142.46 \pm 0.81_{(\times 1.85)}$ |
| VideoReward [arXiv'25] | 55.07 | 57.91 | 23.15 | 82.21 | 45.67 | $52.80_{(+0.90)}$ | $283.63 \pm 2.42_{(\times 3.67)}$ |
| EvoSearch$^\dagger$ [arXiv'25] | **59.25** | **61.09** | **26.18** | **86.73** | 41.80 | $\underline{55.01}_{(+3.11)}$ | $783.76 \pm 3.15_{(\times 10.15)}$ |
| **LatSearch (Ours)** | $\underline{58.12}$ | 59.37 | 22.69 | 82.59 | $\underline{46.44}$ | $53.84_{(+1.94)}$ | $182.43 \pm 6.53_{(\times 2.36)}$ |
| **LatSearch$^\dagger$ (Ours)** | 57.53 | 57.36 | $\underline{23.96}$ | 84.01 | **53.41** | $\textbf{55.25}_{(+3.35)}$ | $164.41 \pm 4.79_{(\times 2.13)}$ |

Table 2: Comparison of varied search budgets $N$ on VBench-2.0.

| Search Budget $N$ | Creativity | Commonsense | Controllability | Human Fidelity | Physics | Average | Inference Time (s) |
|---|---|---|---|---|---|---|---|
| Baseline | 53.81 | 55.63 | 21.99 | 82.11 | 45.98 | 51.90 | $77.21 \pm 0.26$ |
| 4 | 57.70 | 54.70 | 22.00 | 83.63 | 46.03 | $52.81_{(+0.91)}$ | $132.71 \pm 2.55_{(\times 1.72)}$ |
| 6 | 58.12 | **59.37** | **22.69** | 82.59 | 46.44 | $53.84_{(+1.94)}$ | $182.43 \pm 6.53_{(\times 2.36)}$ |
| 8 | **58.47** | 58.87 | 22.26 | **84.00** | 47.05 | $\textbf{54.13}_{(+2.23)}$ | $225.56 \pm 15.57_{(\times 2.92)}$ |

prompts that are non-overlapping with VBench-2.0. Using these prompts, we generate 5,000 videos with different random seeds, while also storing the corresponding latents at selected timesteps and their similarity to the final denoised latent. The dataset is partitioned into 80% for training and 20% for testing. For latent reward model training, we initialise the learning rate at 1e-4 and reduce it to 1e-5 at the 10th epoch. The model is trained with a batch size of 4 and stopped at the 15th epoch. Both the regression loss and preference loss are weighted equally with a coefficient of 1.0. We adopt Wan2.1-1.3B (Wan et al., 2025) as the baseline video generative model, where inference steps are set to 50 and the CFG scale to 5.0 as suggested in (Wan et al., 2025). Following (He et al., 2025), we use random seed 42 for all experiments, and each generated video consists of 33 frames at a resolution of $832 \times 480$. For LatSearch, the scoring schedule is applied at timesteps 10, 15, and 20. All experiments are conducted on NVIDIA A100 GPUs.

**Evaluation.** To evaluate the performance of each method, we use VBench-2.0 (Zheng et al., 2025), a comprehensive benchmark that automatically evaluates video generative models for their intrinsic faithfulness. Specifically, VBench-2.0 assesses video performance across five emerging dimensions beyond superficial faithfulness: Human Fidelity, Controllability, Creativity, Physics, and Commonsense. The scores range from 0 to 100, with a higher score indicating better performance in the corresponding aspects. For each noise prior, we generate 3,860 videos for VBench-2.0 evaluation. For more details, please refer to the Appendix A.3.

## 4.2 COMPARISONS TO STATE OF THE ART

To validate the effectiveness of our method for video inference-time scaling, we first compare it against existing inference-time scaling approaches on VBench-2.0. Specifically, we evaluate against methods that optimise the initial noise, including FreeInit (Wu et al., 2024) and FreqPrior (Yuan et al., 2025), as well as search-based approaches, including VideoReward (Liu et al., 2025b) and EvoSearch (He et al., 2025). We report results across the five evaluation dimensions of VBench-2.0, along with their averaged scores, and also provide the inference time of each method. The detailed experimental settings for all methods are provided in the Appendix A.3.

The main comparison results are reported in Table 1. For methods that optimise the initial noise, although they do not significantly increase computational cost, the absence of an effective verifier limits their effectiveness: simply extracting temporal features from the latent space and feeding them back into the initial noise fails to improve video generation quality. In contrast, search-based approaches rely on output reward optimisation, either via greedy or evolutionary algorithms. While these methods improve quality, they require several times more search time than the baseline, undermining efficiency. Our approach differs in that it evaluates the latent states directly at intermediate steps and employs a probabilistic sampling strategy to retain promising candidate seeds adaptively. As shown in Table 1, our best configuration improves video quality by 3.35% over the baseline,

Figure 3: Qualitative comparison with search-based video generation methods. VideoReward achieves strong semantic alignment but suffers from poor temporal dynamics. EvoSearch improves both semantics and dynamics, yet requires heavy search cost. Our LatSearch reaches comparable quality to EvoSearch while being nearly $5\times$ faster. Results are better viewed with zoom-in.

Table 3: Video generation results on the Wan2.1-14B backbone.

| Search Budget N | Creativity | Commonsense | Controllability | Human Fidelity | Physics | Average |
|---|---|---|---|---|---|---|
| Baseline | 55.21 | 56.18 | 21.79 | 89.74 | 39.96 | 52.58 |
| LATSEARCH | **56.28** | **57.64** | **22.52** | **91.45** | **40.17** | **53.61**$_{(+1.03)}$ |

Table 4: Comparison of VBench-2.0 results under different search strategies and latent reward model settings. A beam-search–style strategy is used for the variant without our PGRP. PL denotes a latent reward model trained with preference loss.

| Methods | PL | RGRP | Creativity | Commonsense | Controllability | Human Fidelity | Physics | Average | Inference Time (s) |
|---|---|---|---|---|---|---|---|---|---|
| Baseline | $\times$ | $\times$ | 53.81 | 55.63 | 21.79 | 89.74 | 39.96 | 52.58 | 77.21 ($\pm$0.26) |
| | $\checkmark$ | $\times$ | $56.63_{(+2.82)}$ | $57.07_{(+1.44)}$ | $22.34_{(+0.35)}$ | $82.54_{(+0.43)}$ | $43.16_{(-2.82)}$ | $52.35_{(+0.45)}$ | $171.23(\pm1.42)_{(\times2.22)}$ |
| LATSEARCH | $\times$ | $\checkmark$ | $56.44_{(+2.64)}$ | $58.51_{(+2.89)}$ | $22.28_{(+0.29)}$ | $\mathbf{84.33}_{(+2.22)}$ | $45.54_{(-0.45)}$ | $53.42_{(+1.52)}$ | $182.43(\pm6.53)_{(\times2.36)}$ |
| | $\checkmark$ | $\checkmark$ | $\mathbf{58.12}_{(+4.31)}$ | $\mathbf{59.37}_{(+3.74)}$ | $\mathbf{22.69}_{(+0.70)}$ | $82.59_{(+0.48)}$ | $\mathbf{46.44}_{(+0.46)}$ | $\mathbf{53.84}_{(+1.94)}$ | $182.43(\pm6.53)_{(\times2.36)}$ |

while requiring only $2.13\times$ more computation. Compared to FreqPrior, under the same computational budget, our method achieves a 2.44% higher quality score. When compared with EvoSearch, although our method achieves only a modest 0.24% gain in quality, it is $4.77\times$ faster. Qualitative comparisons are presented in Figure 3.

## 4.3 ABLATION ANALYSIS

**Effect of Search Budget** $N$**.** We evaluate how the number of candidate trajectories $N$ influences the performance of LATSEARCH. As reported in Table 2, increasing the search budget leads to consistent improvements across all VBench-2.0 dimensions. Moving from $N = 4$ to $N = 6$ yields noticeable gains, particularly in Creativity, Commonsense, and Human Fidelity, demonstrating that maintaining a larger pool of latent trajectories enables the search mechanism to explore richer solution paths. Further increasing the budget to $N = 8$ continues to improve performance, indicating that the proposed latent-level scoring and resampling procedure can effectively utilise additional candidates when available. However, the benefits gradually saturate with growing $N$. Although $N = 8$ produces the strongest results, its marginal improvement over $N = 6$ is smaller compared to the earlier jump from $N = 4$. Importantly, the increase in computational cost remains moderate because evaluations are performed directly in latent space, avoiding repeated decoding into video space. This property allows LATSEARCH to scale gracefully with search budget while remaining computationally practical.

**Adaptability.** To assess the generality of our approach, we evaluate LATSEARCH on a larger video diffusion backbone. Because the method operates entirely in latent space and does not rely on architecture-specific components, it naturally extends to different model scales. When applied to the Wan2.1-14B backbone, as the results shown in Table 3, LATSEARCH consistently improves video quality across all VBench-2.0 dimensions, demonstrating that latent-level reward guidance remains effective regardless of backbone capacity. These results confirm that the proposed search mechanism is model-agnostic and can adapt reliably to stronger or larger-generation models without modification.

**Effectiveness of the RGRP.** We compare RGRP with a beam search–style approach that relies solely on cumulative reward scores. Concretely, both methods follow the same setting of scoring latent candidates at scheduled denoising timesteps. The difference is that the beam search baseline retains a fixed number of seeds purely based on accumulated rewards, while our RGRP method incorporates probabilistic resampling and pruning guided by reward signals. As shown in Table 4,

RGRP achieves consistently better results across most of the five evaluation dimensions. On average, it improves video quality by 0.42% compared to the beam search–style baseline, while incurring only a marginal increase in computation. These results highlight the benefit of probabilistic selection in balancing exploration and exploitation, leading to more robust search outcomes than deterministic reward accumulation. This demonstrates that RGRP mitigates overfitting to accumulated reward signals and promotes more diverse yet high-quality candidate selection.

**Effectiveness of the Preference Loss.** To validate the effectiveness of incorporating preference loss into the latent reward model, we conduct experiments from two perspectives: (1) the consistency between the latent reward and the video-level verifier, and (2) the improvement in video generation quality under our proposed LATSEARCH framework. For the first perspective, we construct approximately 446K latent pairs in the test set, with preferences computed from the corresponding video scores and similarity to the final clean latent. As shown in Figure 4, compared with regression loss, preference loss improves the alignment accuracy by 3.1%, 2.65%, 2.96%, 3.54%, and 3.21% at denoising steps of 10, 15, 20, 25, and 30, respectively. This demonstrates the effectiveness of preference

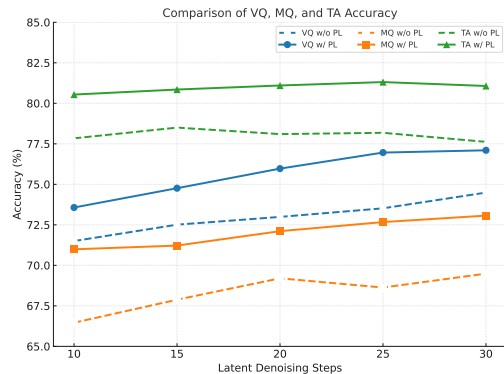

Figure 4: Comparison of VQ, MQ, and TA accuracy across different loss function settings and denoising steps.

learning. Furthermore, we directly validate the preference loss within our latent search method without RGRP. As shown in Table 4, the use of preference loss leads to a 1.07% improvement in video quality. This further highlights the effectiveness of our search algorithm: as the latent reward model becomes stronger, our method consistently achieves better performance. These results confirm that preference-based supervision is not only beneficial at the model level but also translates into measurable improvements in downstream video generation.

## 5 CONCLUSION

We have presented LATSEARCH, a new inference-time scaling framework for video diffusion that addresses the limitations of existing methods, which either impose priors on initial noise or evaluate only final decoded videos. By introducing a latent reward model for intermediate evaluation and combining it with Reward-Guided Resampling and Pruning (RGRP), LATSEARCH achieves both higher video quality (2.44% higher fidelity quality when compute cost equals) and greater efficiency (4.77× faster when fidelity quality equals) against states of the art. Experiments on VBench-2.0 confirm consistent improvements across diverse evaluation dimensions, establishing latent reward guidance as a promising direction for scalable and efficient video generation.

## 6 LIMITATIONS & FUTURE WORK

**Limitations.** While LATSEARCH demonstrates consistent performance improvements and considerable inference-time savings, several limitations remain. Firstly, our resampling-and-pruning procedure is inspired by Sequential Monte Carlo methods, yet theoretical convergence guarantees are difficult to establish due to the learned and approximate nature of the latent reward model. Consequently, although empirically effective, we do not claim formal optimality of the search procedure. Secondly, we rely on cosine-similarity weighting to transform video-level rewards into latent-level supervision. Although ablations show this strategy performs best among tested alternatives, it remains an approximation of true semantic contribution. More principled or learned temporal credit-assignment functions may further improve latent-reward consistency.

**Future Work.** Firstly, developing a lightweight temporal similarity estimator—potentially contrastive or self-supervised—could provide a more accurate credit-assignment mechanism, directly addressing the current approximation bottleneck. Secondly, since two dimensions of our reward model (visual quality and motion quality) are modality-agnostic, LATSEARCH can be extended to audio–video generation, video editing, and instruction-guided transformations, by replacing the text-alignment objective with a task-specific alignment module.

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

# A APPENDIX

## A.1 USE OF LLM STATEMENT

In preparing this submission, we made limited use of publicly available large language models (LLMs), such as ChatGPT, solely for language refinement. This included grammar correction, sentence rephrasing, and improving clarity of exposition. No LLMs were used to generate research ideas, design methodologies, conduct experiments, or produce results. All technical contributions, implementations, and analyses presented in this paper are entirely the work of the authors.

## A.2 ALGORITHMS

---

**Algorithm 1:** Training the Latent Reward Model

---

**Input:** Training set $\{(z, p, r^{\mathrm{VQ}}, r^{\mathrm{MQ}}, r^{\mathrm{TA}}, s_t, t)\}$ (latent tensors $z$, prompt $p$, reward labels $r^d$, latent similarity $s$, denoising step $t$). Model $R_\psi$; optimizer $\mathcal{O}$; loss weights $\lambda_{\mathrm{reg}}^d, \lambda_{\mathrm{pref}}^d$.
**Output:** Updated reward model parameters $\psi$.
**for** *epoch* $= 1 \ldots E$ **do**
 **for** *batch* $\in$ *dataloader* **do**
  $\hat{r} = R_\psi(z, p, t)$          // Predict $[\hat{r}^{\mathrm{VQ}}, \hat{r}^{\mathrm{MQ}}, \hat{r}^{\mathrm{TA}}]$
  **for** *each dimension* $d$ **do**
   $\mathcal{L}_{\mathrm{reg}}^d \leftarrow \|\hat{r}^d - r^d\|^2$        // Regression loss
   For all pairs $(i, j)$:
    $\Delta \hat{r}_{ij}^d \leftarrow \hat{r}_i^d - \hat{r}_j^d, \quad y_{ij}^d \leftarrow \mathbb{I}[r_i^d > r_j^d]$
    $\mathcal{L}_{\mathrm{pref}}^d \leftarrow \mathrm{BCEWithLogits}(\Delta \hat{r}_{ij}^d, y_{ij}^d)$    // Preference loss
  $\mathcal{L} \leftarrow \sum_d \left( \lambda_{\mathrm{reg}}^d \mathcal{L}_{\mathrm{reg}}^d + \lambda_{\mathrm{pref}}^d \mathcal{L}_{\mathrm{pref}}^d \right)$     // Combined loss
  $\mathcal{O}.zero\_grad()$; Backpropagate $\nabla_\psi \mathcal{L}$; $\mathcal{O}.step()$    // Optimization

---

**Algorithm 2:** LatSearch: Latent Reward-Guided Inference Time Search

---

**Input:** Prompt $p$; frame number $F$; resolution $(H, W)$; sampling steps $T$; guidance scale $w$; search schedule $\mathcal{S}$; number of candidates $N$; noise mixing $\eta$; reward verifier $\hat{\mathcal{R}}_\psi$.
**Output:** Generated video $\mathbf{x}$.
**Initialization:**
Sample base noise $z_T^{(0)} \sim \mathcal{N}(0, I)$; sample perturbations $\epsilon_i \sim \mathcal{N}(0, I)$ for $i = 1, \ldots, N-1$;
Construct candidate set $z_T^{(i)} \leftarrow \sqrt{1 - \eta^2} \, z_T^{(0)} + \eta \, \epsilon_i$;
Initialize weights $w_i \leftarrow 1/N$ and cumulative weights $c_i \leftarrow 0$;
Instantiate $N$ independent diffusion schedulers.

**for** $j = 1$ **to** $T$ **do**
 **for** $i = 1$ **to** $N$ **do**
  $\epsilon_\theta \leftarrow \epsilon_\theta(z_t, t, \varnothing) + w[\epsilon_\theta(z_t, t, c) - \epsilon_\theta(z_t, t, \varnothing)]$ // Classifier-free guidance
  $z_{t_{j-1}}^{(i)} \leftarrow \mathrm{SamplerStep}(z_{t_j}^{(i)}, \hat{\epsilon})$       // One sampler step
 // Evaluate and resample candidates
 **if** $j \in \mathcal{S}$ **then**
  $r_i \leftarrow \mathcal{R}(z_{t_{j-1}}^{(i)}, p, t_j)$ for $i = 1, \ldots, N$
  $w_i \leftarrow \mathrm{Softmax}(r_i); \quad c_i \leftarrow c_i + w_i$
  Resample indices $\mathcal{I} \sim \mathrm{Multinomial}(w)$
  Retain unique $\{z^{(i)}, c_i\}_{i \in \mathcal{I}}$ and their schedulers; set $N \leftarrow |\mathcal{I}|$
  // Final pruning
  **if** $j = \max(\mathcal{S})$ **then**
   $i^\star \leftarrow \arg\max_i c_i$; retain only $z^{(i^\star)}$ and scheduler; set $N \leftarrow 1$

**Decode:** $\mathbf{x} \leftarrow \mathrm{VAE.decode}(z_0)$
**return** $\mathbf{x}$

---

## A.3 Detailed Experimental Settings

**Calculation of VBench-2.0 Matrices.** We report evaluation results based on the VBench-2.0 metrics. Specifically, each high-level score is computed as the mean of several fine-grained dimensions:

- **Creativity** Score = average of Diversity and Composition.
- **Commonsense** Score = average of Motion Rationality and Instance Preservation.
- **Controllability** Score = average of Dynamic Spatial Relationship, Dynamic Attribute, Motion Order Understanding, Human Interaction, Complex Landscape, Complex Plot, and Camera Motion.
- **Human Fidelity** Score = average of Human Anatomy, Human Identity, and Human Clothes.
- **Physics** Score = average of Mechanics, Thermotics, Material, and Multi-View Consistency.

Finally, the Total Score is obtained by averaging the above five high-level scores: Creativity, Commonsense, Controllability, Human Fidelity, and Physics.

**Implementations of Compared Methods.** We compare our approach with two types of baselines: (i) noise-optimisation methods, including FreeInit (Wu et al., 2024) and FreqPrior (Yuan et al., 2025), and (ii) search-based methods, including VideoReward (Liu et al., 2025b) and EvoSearch (He et al., 2025). All methods use a random seed of 42.

- **FreeInit**(Wu et al., 2024): We follow the original setup, using 4 extra sampling iterations and applying a Butterworth filter with a normalised spatial-temporal cutoff frequency of 0.25 as the low-pass filter.
- **FreqPrior**(Yuan et al., 2025): Following the paper's setting, we use 2 extra sampling iterations and the same Butterworth filter configuration as FreeInit. The timestep $t$ is set to 768, and the ratio $cos\theta$ is set to 0.8.
- **VideoReward** (Liu et al., 2025b): We adopt a best-of-N search strategy. Specifically, we sample 4 different initial noise tensors, denoise each through the full trajectory, and decode them into videos. The video reward model is then used to evaluate video quality, motion quality, and text alignment, after which we select the highest-scoring video as the final output.
- **EvoSearch** (He et al., 2025): We follow the original configuration, setting the population size schedule to {6, 3, 3} and the evolution schedule to {5, 20}. We use the DPM++ solver, with an elite size of 3 and a mutation rate of 0.2.

## A.4 Additional Experimental Results

In this section, we provide additional experimental results to further validate the latent reward model, the search procedure, and the runtime efficiency of our method.

### A.4.1 Effect of Preference Loss

Table 5: Comparison of VQ, MQ, and TA accuracy across different loss function settings and denoising steps. PL denotes preference loss.

| Latent Denoising Steps | VQ Accuracy | | MQ Accuracy | | TA Accuracy | | Average Accuracy | |
|---|---|---|---|---|---|---|---|---|
| | w/o PL | w/ PL | w/o PL | w/ PL | w/o PL | w/ PL | w/o PL | w/ PL |
| 10 | 71.50 | **73.57** | 66.46 | **70.99** | 77.84 | **80.54** | 71.93 | **75.03** |
| 15 | 72.51 | **74.76** | 67.88 | **71.22** | 78.50 | **80.85** | 72.96 | **75.61** |
| 20 | 72.99 | **75.97** | 69.20 | **72.11** | 78.10 | **81.10** | 73.43 | **76.39** |
| 25 | 73.52 | **76.96** | 68.63 | **72.67** | 78.18 | **81.31** | 73.44 | **76.98** |
| 30 | 74.49 | **77.10** | 69.49 | **73.07** | 77.62 | **81.07** | 73.87 | **77.08** |

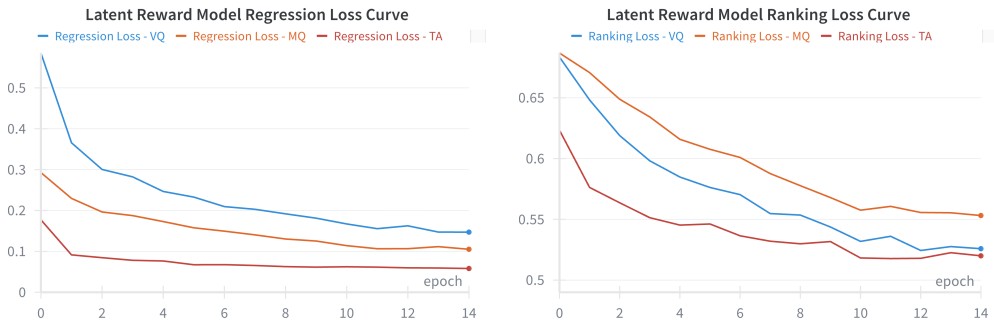

Figure 5: Training curves of the latent reward model: regression loss and reference loss.

### A.4.2 TRAINING CURVES OF THE LATENT REWARD MODEL

### A.4.3 FULL VBENCH-2.0 RESULTS ACROSS ALL METHODS AND BACKBONES

Table 6: VBench-2.0 evaluation results per dimension across different methods. [†] indicates the use of DPM-Solver++.

| Methods | Diversity | Composition | Motion Rationality | Instance Preservation | Dynamic Spatial Relationship | Dynamic Attribute |
|---|---|---|---|---|---|---|
| Baseline (Wan2.1-1.3B) | 65.85 | 41.76 | 25.29 | 85.96 | 30.92 | 11.36 |
| + FreeInit | 55.49 | 38.11 | 27.01 | **91.81** | 29.95 | 9.16 |
| + FreqPrior | 66.35 | 41.04 | 27.01 | 84.21 | 28.50 | 6.59 |
| + VideoReward | 62.74 | 47.40 | 31.61 | 84.21 | 31.88 | 13.92 |
| + EvoSearch[†] | **71.88** | 46.62 | **35.05** | 87.13 | **35.75** | 11.72 |
| + LATSEARCH (w/o RGRP) | 66.18 | 46.70 | 29.89 | 87.13 | 33.82 | 13.92 |
| + LATSEARCH | 68.33 | **47.90** | 32.76 | 85.97 | 29.95 | **15.38** |
| + LATSEARCH[†] | 69.19 | 45.87 | 28.16 | 86.55 | 29.95 | 13.55 |
| Baseline (Wan2.1-14B) | 62.34 | **48.08** | 28.73 | 83.62 | 24.15 | **13.55** |
| + LATSEARCH | **66.18** | 46.39 | **29.31** | **85.97** | **25.60** | 13.19 |

| Methods | Motion Order Understanding | Human Interaction | Complex Landscape | Complex Plot | Camera Motion | Human Anatomy |
|---|---|---|---|---|---|---|
| Baseline | 16.84 | 54.00 | 16.22 | 11.33 | 13.27 | 80.67 |
| + FreeInit | 17.17 | 53.00 | 18.00 | **11.85** | 15.12 | 81.53 |
| + FreqPrior | 16.50 | 48.33 | **21.33** | 11.63 | 9.57 | 79.61 |
| + VideoReward | 17.51 | 54.33 | 18.22 | 11.69 | 14.51 | 78.93 |
| + EvoSearch[†] | **21.88** | 65.67 | 19.78 | 9.94 | **18.52** | **82.24** |
| + LATSEARCH (w/o RGRP) | 18.86 | 48.67 | 17.78 | 9.31 | 13.58 | 79.54 |
| + LATSEARCH | 20.88 | 50.67 | 17.56 | 11.44 | 12.96 | 77.97 |
| + LATSEARCH[†] | 19.87 | 57.33 | 17.78 | 10.69 | **18.52** | 79.62 |
| Baseline (Wan2.1-14B) | **22.29** | 55.33 | 14.44 | **12.90** | 9.87 | 87.24 |
| + LATSEARCH | 18.58 | **61.67** | **16.22** | 12.17 | **10.19** | 89.53 |

| Methods | Human Identity | Human Clothes | Mechanics | Thermotics | Material | Multi-View Consistency |
|---|---|---|---|---|---|---|
| Baseline | 68.41 | 97.24 | 60.33 | 55.47 | 32.84 | 35.28 |
| + FreeInit | 75.22 | **98.93** | 52.14 | 44.48 | 32.47 | 13.52 |
| + FreqPrior | 73.55 | 97.69 | 53.09 | 53.03 | 27.27 | 21.04 |
| + VideoReward | 71.33 | 96.39 | 60.17 | **57.97** | 33.33 | 31.24 |
| + EvoSearch[†] | **80.19** | 97.77 | 54.76 | 47.65 | 35.16 | 29.63 |
| + LATSEARCH (w/o RGRP) | 76.15 | 97.30 | 53.97 | 51.52 | **40.26** | 36.39 |
| + LATSEARCH | 74.37 | 95.43 | **62.50** | 50.37 | 37.97 | 34.93 |
| + LATSEARCH[†] | 74.22 | 98.18 | 61.72 | 46.57 | 37.50 | **67.85** |
| Baseline (Wan2.1-14B) | 83.82 | 98.16 | **52.17** | **47.88** | 36.20 | **23.60** |
| + LATSEARCH | **85.28** | **99.54** | 52.03 | 47.14 | **38.81** | 22.71 |

Table 7: Impact of search budget, temperature, and search schedule on VBench-2.0 performance across all dimensions.

| Methods | Diversity | Composition | Motion Rationality | Instance Preservation | Dynamic Spatial Relationship | Dynamic Attribute |
|---|---|---|---|---|---|---|
| Baseline | 65.85 | 41.76 | 25.29 | 85.96 | 30.92 | 11.36 |
| Search Budget $N = 4$ | 70.32 | 45.08 | 31.03 | 78.36 | 31.41 | 15.02 |
| Search Budget $N = 6$ | 68.33 | 47.90 | 32.76 | 85.97 | 29.95 | 15.38 |
| Search Budget $N = 8$ | 69.19 | 47.75 | 31.61 | 86.12 | 29.47 | 12.46 |
| Temperature $\tau = 0.5$ | 66.99 | 46.49 | 29.89 | 86.61 | 28.50 | 13.92 |
| Temperature $\tau = 1.0$ | 68.33 | 47.90 | 32.76 | 85.97 | 29.95 | 15.38 |
| Temperature $\tau = 2.0$ | 69.19 | 47.64 | 31.61 | 84.85 | 24.15 | 15.38 |
| $\{10, 15\}$ | 69.07 | 46.94 | 31.61 | 80.70 | 30.43 | 15.75 |
| $\{10, 15, 20\}$ | 68.33 | 47.90 | 32.76 | 85.97 | 29.95 | 15.38 |
| $\{10, 15, 20, 25\}$ | 68.13 | 46.46 | 28.16 | 85.44 | 32.85 | 17.95 |
| $\{10, 15, 20, 25, 30\}$ | 65.23 | 45.57 | 29.31 | 85.95 | 29.47 | 15.38 |

| Methods | Motion Order Understanding | Human Interaction | Complex Landscape | Complex Plot | Camera Motion | Human Anatomy |
|---|---|---|---|---|---|---|
| Baseline | 16.84 | 54.00 | 16.22 | 11.33 | 13.27 | 80.67 |
| Search Budget $N = 4$ | 19.87 | 48.33 | 18.00 | 10.23 | 11.11 | 78.42 |
| Search Budget $N = 6$ | 20.88 | 50.67 | 17.56 | 11.44 | 12.96 | 77.97 |
| Search Budget $N = 8$ | 20.51 | 50.67 | 18.21 | 9.65 | 14.82 | 78.51 |
| Temperature $\tau = 0.5$ | 18.86 | 51.67 | 17.11 | 9.14 | 13.58 | 78.18 |
| Temperature $\tau = 1.0$ | 20.88 | 50.67 | 17.56 | 11.44 | 12.96 | 77.97 |
| Temperature $\tau = 2.0$ | 17.51 | 51.67 | 17.56 | 10.12 | 14.51 | 79.31 |
| $\{10, 15\}$ | 17.85 | 49.00 | 17.11 | 9.01 | 15.43 | 78.92 |
| $\{10, 15, 20\}$ | 20.88 | 50.67 | 17.56 | 11.44 | 12.96 | 77.97 |
| $\{10, 15, 20, 25\}$ | 14.82 | 50.33 | 17.33 | 10.54 | 14.81 | 78.66 |
| $\{10, 15, 20, 25, 30\}$ | 15.82 | 49.00 | 16.22 | 10.44 | 14.81 | 77.73 |

| Methods | Human Identity | Human Clothes | Mechanics | Thermotics | Material | Multi-View Consistency |
|---|---|---|---|---|---|---|
| Baseline | 68.41 | 97.24 | 60.33 | 55.47 | 32.84 | 35.28 |
| Search Budget $N = 4$ | 76.51 | 95.95 | 56.57 | 52.24 | 41.56 | 33.76 |
| Search Budget $N = 6$ | 74.37 | 95.43 | 62.50 | 50.37 | 37.97 | 34.93 |
| Search Budget $N = 8$ | 77.10 | 96.38 | 62.11 | 56.55 | 35.23 | 34.29 |
| Temperature $\tau = 0.5$ | 79.04 | 96.38 | 56.67 | 51.49 | 38.96 | 38.64 |
| Temperature $\tau = 1.0$ | 74.37 | 95.43 | 62.50 | 50.37 | 37.97 | 34.93 |
| Temperature $\tau = 2.0$ | 72.95 | 94.09 | 58.54 | 51.13 | 36.14 | 39.46 |
| $\{10, 15\}$ | 73.17 | 95.05 | 59.50 | 49.21 | 38.96 | 21.20 |
| $\{10, 15, 20\}$ | 74.37 | 95.43 | 62.50 | 50.37 | 37.97 | 34.93 |
| $\{10, 15, 20, 25\}$ | 72.61 | 96.86 | 60.80 | 52.59 | 35.44 | 37.61 |
| $\{10, 15, 20, 25, 30\}$ | 73.33 | 94.09 | 61.75 | 50.56 | 36.71 | 32.34 |

### A.4.4 SENSITIVITY TO SEARCH HYPERPARAMETERS

We analyse the robustness of LATSEARCH with respect to three key hyperparameters.

*(i) Search Budget $N$.* Increasing $N$ enhances exploration but raises compute cost. As shown in Table 7, performance improves monotonically from $N=4$ to $N=8$, confirming that the search mechanism scales reliably with additional candidates.

*(ii) Temperature $\tau$.* Varying $\tau \in \{0.5, 1.0, 2.0\}$ leads to only small changes in VBench-2.0 scores (Table 7), indicating that the resampling step is stable across a wide temperature range.

*(iii) Scoring-timestep schedule $S$.* The choice of scoring timesteps is crucial because reward quality varies substantially across the diffusion trajectory. We avoid applying scoring at very early steps because latents in these stages are dominated by noise and contain almost no semantic content, making reward estimation unstable and non-informative.

Conversely, we do not apply scoring at very late timesteps primarily due to computational cost. For instance, with $N=6$ candidates and a DiT forward time of 1.35 s per step, scoring within the

Table 8: Effect of scoring timestep schedules on VBench-2.0.

| Search Scheduler | Creativity | Commonsense | Controllability | Human Fidelity | Physics | Average | Inference Time (s) |
|---|---|---|---|---|---|---|---|
| Baseline | 53.81 | 55.63 | 21.99 | 82.11 | 45.98 | **51.90** | $77.21 \pm 0.26$ |
| {10, 15} | 58.01 | 56.16 | 22.08 | 82.38 | 42.22 | **52.17 (+0.27)** | $154.63 \pm 3.06$ |
| {10, 15, 20} | 58.12 | 59.37 | 22.69 | 82.59 | 46.44 | **53.84 (+1.94)** | $182.43 \pm 6.53$ |
| {10, 15, 20, 25} | 57.30 | 56.80 | 22.66 | 82.71 | 46.61 | **53.22 (+1.32)** | $186.47 \pm 8.58$ |
| {10, 15, 20, 25, 30} | 55.40 | 57.63 | 21.59 | 81.72 | 45.34 | **52.34 (+0.44)** | $198.48 \pm 10.99$ |

Table 9: Comparison of VQ, MQ, and TA accuracy across different credit assignment strategies and denoising steps.

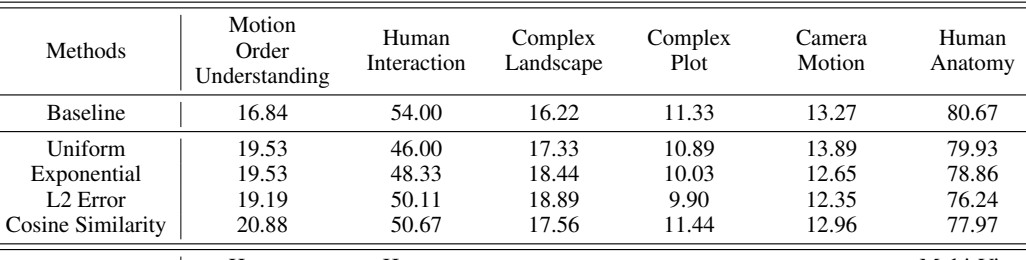

| Steps | VQ Accuracy | | | | MQ Accuracy | | | | TA Accuracy | | | | Average Accuracy | | | |
|---|---|---|---|---|---|---|---|---|---|---|---|---|---|---|---|---|
| | Uni. | Exp. | L2 | Cos. | Uni. | Exp. | L2 | Cos. | Uni. | Exp. | L2 | Cos. | Uni. | Exp. | L2 | Cos. |
| 10 | 72.48 | 72.72 | 74.81 | 73.57 | 67.36 | 68.27 | 68.68 | 70.99 | 81.15 | 80.57 | 80.56 | 80.54 | 73.66 | 73.85 | 74.68 | 75.03 |
| 15 | 73.56 | 73.94 | 75.20 | 74.76 | 69.40 | 69.84 | 70.34 | 71.22 | 81.26 | 80.66 | 81.22 | 80.85 | 74.74 | 74.81 | 75.59 | 75.61 |
| 20 | 73.85 | 73.13 | 75.56 | 75.97 | 69.38 | 69.15 | 70.69 | 72.11 | 81.46 | 81.25 | 81.13 | 81.10 | 74.90 | 74.51 | 75.79 | 76.39 |
| 25 | 73.88 | 73.42 | 76.07 | 76.96 | 68.86 | 69.36 | 70.74 | 72.67 | 81.36 | 81.19 | 81.47 | 81.31 | 74.70 | 74.66 | 76.09 | 76.98 |
| 30 | 74.27 | 73.77 | 76.76 | 77.10 | 68.63 | 69.82 | 70.72 | 73.07 | 81.27 | 81.34 | 81.58 | 81.07 | 74.72 | 74.98 | 76.35 | 77.08 |
| Average | 73.61 | 73.40 | **75.68** | 75.67 | 68.73 | 69.29 | 70.23 | **72.01** | 81.30 | 81.00 | **81.19** | 80.97 | 74.54 | 74.56 | 75.70 | **76.22** |

Table 10: VBench-2.0 evaluation results per dimension under different credit assignment strategies.

| Methods | Diversity | Composition | Motion Rationality | Instance Preservation | Dynamic Spatial Relationship | Dynamic Attribute |
|---|---|---|---|---|---|---|
| Baseline | 65.85 | 41.76 | 25.29 | 85.96 | 30.92 | 11.36 |
| Uniform | 68.49 | 41.46 | 27.01 | 77.78 | 30.72 | 11.72 |
| Exponential | 67.68 | 41.31 | 28.74 | 75.44 | 29.95 | 12.45 |
| L2 Error | 66.65 | 44.90 | 29.59 | 83.68 | 29.57 | 15.38 |
| Cosine Similarity | 68.33 | 47.90 | 32.76 | 85.97 | 29.95 | 15.38 |

| Methods | Motion Order Understanding | Human Interaction | Complex Landscape | Complex Plot | Camera Motion | Human Anatomy |
|---|---|---|---|---|---|---|
| Baseline | 16.84 | 54.00 | 16.22 | 11.33 | 13.27 | 80.67 |
| Uniform | 19.53 | 46.00 | 17.33 | 10.89 | 13.89 | 79.93 |
| Exponential | 19.53 | 48.33 | 18.44 | 10.03 | 12.65 | 78.86 |
| L2 Error | 19.19 | 50.11 | 18.89 | 9.90 | 12.35 | 76.24 |
| Cosine Similarity | 20.88 | 50.67 | 17.56 | 11.44 | 12.96 | 77.97 |

| Methods | Human Identity | Human Clothes | Mechanics | Thermotics | Material | Multi-View Consistency |
|---|---|---|---|---|---|---|
| Baseline | 68.41 | 97.24 | 60.33 | 55.47 | 32.84 | 35.28 |
| Uniform | 77.73 | 97.27 | 55.91 | 58.33 | 43.33 | 33.43 |
| Exponential | 74.79 | 95.45 | 56.91 | 56.55 | 43.04 | 40.68 |
| L2 Error | 75.09 | 95.02 | 64.52 | 54.07 | 36.49 | 32.71 |
| Cosine Similarity | 74.37 | 95.43 | 62.50 | 50.37 | 37.97 | 34.93 |

[10,20] interval increases total inference time by approximately 67.5–135 s (best–worst case). Scoring within [30,40], however, increases the cost to 202–270 s, resulting in a 2–3 times runtime increase. This conflicts with our motivation of enabling early, efficient search.

For this reason, we intentionally apply search as early as possible once semantically meaningful structure emerges. Our ablations in Table 8 confirm this design: too few scoring points (e.g., {10, 15}) provide insufficient temporal coverage, while too many (e.g., {10, 15, 20, 25, 30}) accumulate uncertainty from similarity-derived targets and degrade performance. A moderate, well-spaced mid-range schedule, {10, 15, 20}, achieves the best trade-off between discriminative power and computational efficiency.

### A.4.5 CREDIT ASSIGNMENT STRATEGIES

Tables 9 and 10 evaluate the effect of different latent credit-assignment strategies on both reward-prediction accuracy and final video quality. Cosine-similarity and L2-error weighting achieve the

Table 11: Inference time comparison across different methods.

| Methods | DiT Time | Decoder Time | Reward Time | Total Time | VBench-2.0 Results |
|---|---|---|---|---|---|
| Baseline | $67.46 \pm 0.71$ | $1.85 \pm 0.26$ | 0 | $69.31 \pm 0.76$ | 51.82 |
| VideoReward | $266.71 \pm 0.40$ | $10.12 \pm 0.54$ | $0.56 \pm 0.33$ | $277.39 \pm 0.75$ | 52.80 |
| EvoSearch | $756.66 \pm 1.27$ | $31.99 \pm 0.96$ | $1.92 \pm 1.04$ | $790.57 \pm 1.90$ | 55.01 |
| LatSearch | $156.11 \pm 5.11$ | $1.88 \pm 0.24$ | $1.03 \pm 0.06$ | $159.02 \pm 5.12$ | 55.25 |

Table 12: Runtime of each module.

| Module | Inference Time (sec.) |
|---|---|
| DiT Forward | $1.35 \pm 0.13$ (per step) |
| VAE Decoder | $1.85 \pm 0.26$ (per video) |
| Latent Reward Model | $0.84 \pm 0.003$ (per latent) |

Table 13: Human pairwise preference rates.

| Criterion | Preference for LatSearch |
|---|---|
| Visual Quality | 72.15% |
| Motion Quality | 75.29% |
| Video-Text Alignment | 78.51% |
| **Average** | **75.32%** |

highest average VQ/MQ/TA accuracy, and cosine-based assignment further yields the strongest VBench-2.0 performance across most dimensions, confirming that it provides the most stable and discriminative latent supervision.

### A.4.6 RUNTIME BREAKDOWN

Table 11 provides a complete runtime across DiT, decoder, and reward evaluation. LATSEARCH introduces minimal overhead and achieves the best VBench-2.0 score among all methods with significantly lower total latency than search-based baselines.

We also present the latency of each module in Table 12. The results show that VAE decoding is substantially more expensive than a single DiT forward step, and repeated decoding—required by methods such as VideoReward and EvoSearch—quickly becomes the dominant inference cost. In contrast, LatSearch performs no decoding during search and evaluates the reward model only at a small number of sparse timesteps, keeping the overall overhead minimal.

### A.4.7 HUMAN PREFERENCE STUDY

To further validate the perceptual quality of EVOSEARCH beyond automated VBench-2.0 metrics, we conducted a pairwise human preference study. Specifically, we sampled 40 video pairs generated by the baseline model and EVOSEARCH across diverse categories, including animals, humans, natural scenery, and architecture. Each pair was evaluated by 30 participants, who were asked to choose which video was better along three key criteria:

Visual Quality — clarity, level of detail, and absence of artifacts; Motion Quality — temporal consistency, smoothness, and physical realism; Video–Text Alignment — fidelity of subjects, actions, and scenes to the prompt.

Results in Table 13 indicate that participants expressed a clear preference for EVOSEARCH across all criteria, demonstrating that the improvements observed in automated metrics also align with human subjective judgment.

### A.4.8 MSE ANALYSIS OF LATENT-REWARD VS. DECODED-REWARD PREDICTIONS

To assess how accurately intermediate latents reflect the final video quality, we compute the mean squared error (MSE) between reward estimates at intermediate timesteps and the final video reward. Specifically, for each timestep $t \in \{0, 5, 10, \ldots, 40\}$, we evaluate:

- **Latent-reward prediction** $R_t^{\text{latent}}$ produced directly from the latent reward model.
- **Decoded-intermediate reward** $R_t^{\text{decoded}}$ obtained by decoding $z_t$ using the VAE and applying the full video-level reward model.
- **Final reward** $R_{\text{final}}$ computed by completing the denoising trajectory from $z_t$ to $z_0$, decoding the final video, and scoring it.

For each method, the prediction error is measured as:

$$\text{MSE}(t) = \|R_t - R_{\text{final}}\|_2^2.$$

Figure 6 reports the MSE curves across all VBench-2.0 dimensions.

**Observations.**

1. **Latent-reward predictions exhibit consistently lower error.** In every category and at every timestep, latent-space MSE is substantially smaller than decoded-intermediate MSE, often by a factor of 3–7 times.
2. **Latent MSE decreases smoothly over timesteps.** As denoising progresses, latent features become increasingly structured, and the latent reward model aligns more closely with the eventual video quality.
3. **Decoded-intermediate MSE remains high until very late stages.** Intermediate decoded frames contain artifacts and incomplete semantics, making video-level evaluation unstable and weakly correlated with the final reward.

**Conclusion.** These results show that latent representations carry more predictive semantic information about the final video quality than partially decoded frames. This supports the use of latent-space evaluation for efficient inference-time search.

### A.4.9 QUALITATIVE COMPARISONS.

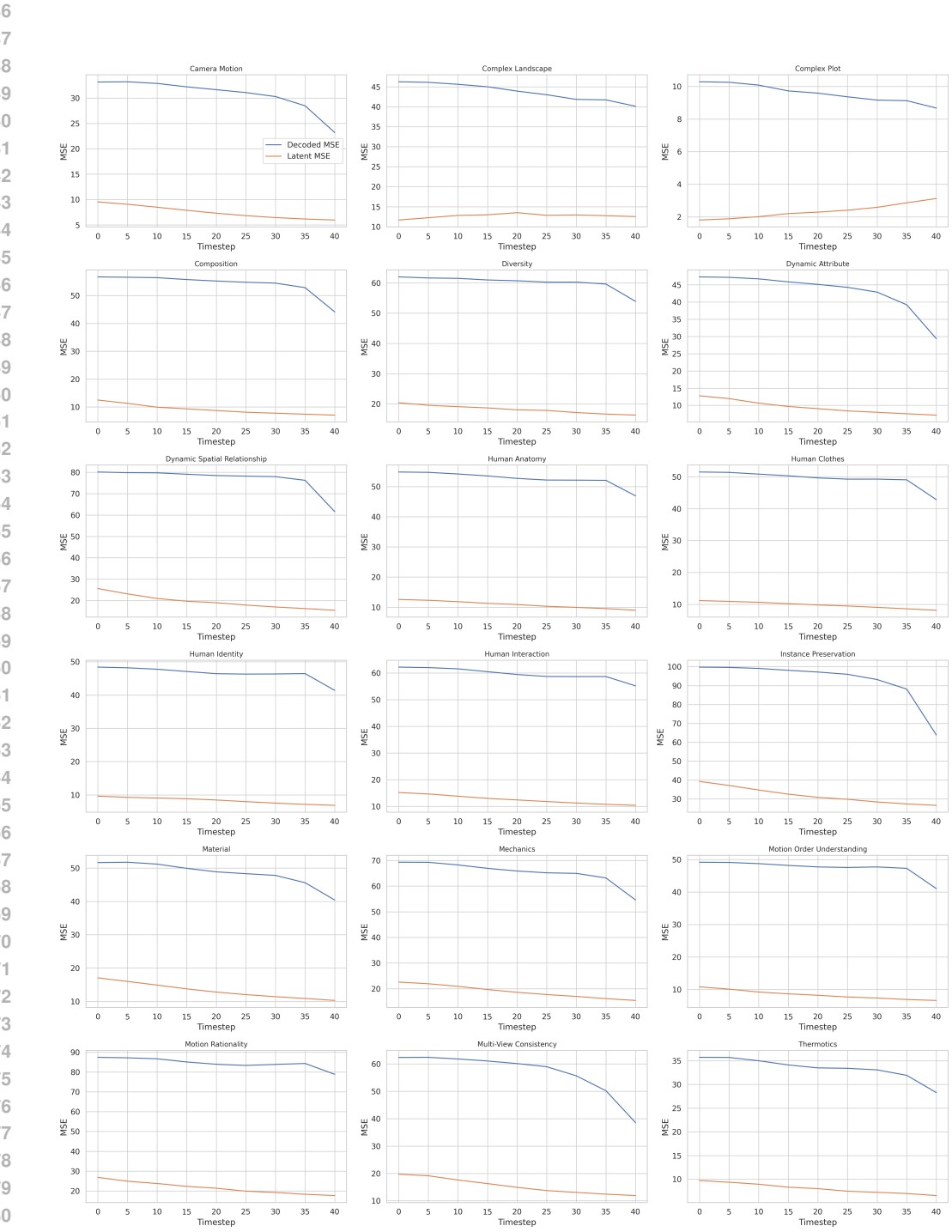

Figure 6: MSE between latent-reward predictions (orange) and decoded-intermediate rewards (blue) across timesteps for all VBench-2.0 dimensions.

**Prompt: A bear with the antlers of a deer, roaming the forest with a regal presence.**

**Prompt: A clear glass of oil is gently poured into a glass of milk.**

**Prompt: A cloud changes from small to large as it gathers moisture.**

Figure 7: Comparison of text-to-video generation results between the baseline model (top) and LATSEARCH (bottom) for each prompt.

**Prompt: A dog is on the right of a sofa, then the dog runs to the front of the sofa.**

**Prompt: A lion with the wings of an eagle, soaring through the sky with majestic ease.**

**Prompt: A man is riding a bike.**

Figure 8: Comparison of text-to-video generation results between the baseline model (top) and LATSEARCH (bottom) for each prompt.

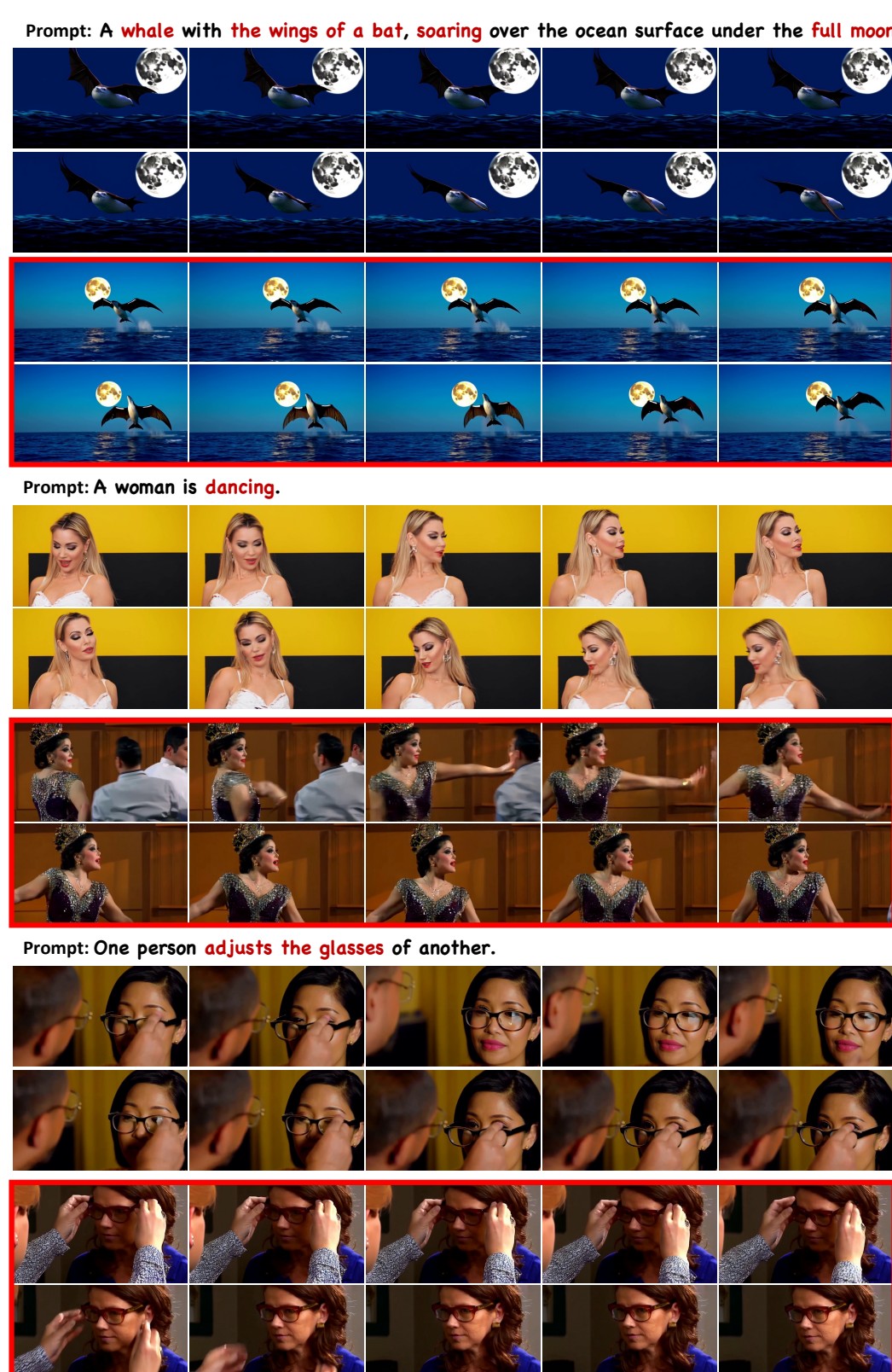

Figure 9: Comparison of text-to-video generation results between the baseline model (top) and LATSEARCH (bottom) for each prompt.

