# OpenReview forum: "Latent Reward-Guided Search for Faster Inference-Time Scaling in Video Diffusion"
_ICLR.cc/2026/Conference — Submitted to ICLR 2026_

### Official Review · Reviewer_ubZH · 2025-10-25

**Soundness:** 2
**Presentation:** 3
**Contribution:** 2
**Rating:** 4
**Confidence:** 3

**Summary:**

This paper presents LATSEARCH, an inference-time scaling framework for video diffusion models, aiming to address limitations of existing methods that either rely on initial noise priors or only evaluate final decoded videos. It introduces a latent reward model that assesses partially denoised latents at arbitrary timesteps across visual quality, motion quality, and text alignment, and integrates this model with a Reward-Guided Resampling and Pruning (RGRP) mechanism to enable efficient search during denoising. Experiments on the VBench-2.0 benchmark show that LATSEARCH consistently improves video generation quality across multiple dimensions compared to the baseline Wan2.1 model.

**Strengths:**

1. The paper introduces a latent reward model capable of evaluating partially denoised latents at arbitrary timesteps to provide intermediate feedback on visual quality, motion quality, and text alignment, addressing the issue of delayed/sparse rewards in existing video diffusion inference methods.
2. The paper proposes the Reward-Guided Resampling and Pruning (RGRP) mechanism: it probabilistically resamples candidates based on reward-normalized weights and prunes to retain the candidate with the highest cumulative reward at the final step, enabling efficient search without wasting computation on low-quality trajectories.
3. Achieves a superior quality-efficiency balance: on VBench-2.0, it improves video generation across key dimensions and matches or exceeds state-of-the-art quality while reducing runtime by up to 79%.

**Weaknesses:**

1. Lacks in-depth theoretical analysis of LATSEARCH’s convergence and optimality, with the work being primarily empirical, which limits the understanding of why the proposed mechanisms (e.g., latent reward guidance, RGRP) work and their generalizability to other video diffusion frameworks.
2. The latent reward model’s similarity-grounded credit assignment (SGCA) relies on cosine similarity to map video-level rewards to intermediate latents, but the choice of cosine similarity is not justified, and no alternative similarity metrics (e.g., Euclidean distance) are tested, raising doubts about the robustness of reward allocation.
3. Key parameters in LATSEARCH (e.g., number of candidates N=4/6, scoring timesteps 10/15/20, temperature τ for resampling) lack sufficient sensitivity analysis; the paper does not explain why these specific values are chosen over others, reducing the reproducibility and persuasiveness of the results.
4. When comparing with state-of-the-art methods (e.g., EvoSearch), it does not verify if the parameters of these baseline methods are optimized for the Wan2.1-1.3B model (they are directly adopted from original papers), potentially leading to unfair comparisons that overstate LATSEARCH’s advantages.
5. The inference time measurement does not explicitly clarify whether it includes the computational cost of the latent reward model (e.g., Qwen2-VL-3B-based scoring), which may underestimate the actual computational overhead and distort the perceived efficiency advantage.

**Questions:**

see weaknesses

---

> ### Author Response · Authors · 2025-11-20
> **Official Comment by Authors [1/3]**
>
> Thank you for your positive evaluation of our contributions in latent-level evaluation and efficient search. We appreciate your thorough and thoughtful feedback. We address each of your comments individually below.
>
> - **Q1. Lacks in-depth theoretical analysis of method’s convergence and optimality.**
>
> **A1.** We thank the reviewer for the insightful comment. We clarify that our objective is to design an inference-time candidate search mechanism for diffusion models under limited compute. Because both the diffusion dynamics and the latent reward model are non-convex and learned, we do not claim global optimality or formal convergence guarantees. We agree that establishing convergence for such a procedure is inherently non-trivial. While our resampling–pruning procedure is conceptually inspired by Sequential Monte Carlo (SMC) methods, and prior work on adaptive SMC methods provides theoretical convergence analyses, including WLLN and CLT results for adaptive resampling and proposal adaptation (Beskos et al., 2014), these analyses rely on exact likelihood-based importance weights and analytically defined target distributions. In contrast, our latent reward model produces learned and approximate credit estimates for intermediate denoising states, which do not satisfy the assumptions required by existing SMC theory. Therefore, providing formal convergence guarantees is outside the scope of this work, and we will explicitly clarify this limitation in the revised paper.
>
> [1]. Beskos A, Jasra A, Kantas N, Thiery A. On the Convergence of Adaptive Sequential Monte Carlo Methods. The Annals of Applied Probability. 2014.
>
> - **Q2. Choice of cosine similarity is not justified. No alternative metrics (e.g., Euclidean distance) are tested.**
>
> **A2.** We thank the reviewer for raising this point. To address it, we provide a comprehensive ablation comparing cosine similarity with several alternative credit-assignment strategies, including L2 distance, uniform weighting, and exponential weighting. As shown in Tables R1 and R2 below, cosine similarity consistently produces the most stable latent supervision and leads to the highest downstream video-generation quality. These results confirm that cosine similarity is not only intuitive but also empirically the most effective among the tested alternatives.
>
> **Table R1. Comparison of VQ, MQ, and TA accuracy across different credits assignment strategies.**
>
> |      Methods      | VQ Accuracy | MQ Accuracy | TA Accuracy |     Average     |
> | :----------------: | :---------: | :---------: | :---------: | :-------------: |
> |      Uniform      |    73.61    |    68.73    |    81.30    | **74.54** |
> |    Exponential    |    73.39    |    69.29    |    81.00    | **74.56** |
> |      L2 Error      |    75.68    |    70.23    |    81.19    | **75.70** |
> | Cosine Similarity |    75.67    |    72.01    |    80.97    | **76.22** |
>
> **Table R2. Generated video quality under different credit assignment strategies.**
>
> |      Methods      | Creativity | Commonsense | Controllability | Human Fidelity | Physics |     Average     |
> | :----------------: | :--------: | :---------: | :-------------: | :------------: | :-----: | :-------------: |
> |      Baseline      |   53.81   |    55.63    |      21.99      |     82.11     |  45.98  | **51.90** |
> |      Uniform      |   54.98   |    52.40    |      21.44      |     84.98     |  47.75  | **52.31** |
> |    Exponential    |   54.50   |    52.09    |      21.63      |     83.03     |  49.30  | **52.11** |
> |      L2 Error      |   55.78   |    56.64    |      22.20      |     82.12     |  46.95  | **52.74** |
> | Cosine Similarity |   58.12   |    59.37    |      22.69      |     82.59     |  46.44  | **53.84** |

---

> ### Author Response · Authors · 2025-11-20
> **Official Comment by Authors [2/3]**
>
> - **Q3. Key parameters (N, timesteps, and temperature) lack sensitivity analysis.**
>
> **A3.** We thank the reviewer for raising this point. We performed additional experiments to examine the robustness of LatSearch with respect to three key factors: (i) the number of candidate trajectories $N$, (ii) the softmax temperature $\tau$, and (iii) the choice of scoring timesteps.
>
> Candidate Number $N$ (Search Budget): Increasing $N$ strengthens exploration but also increases compute. As shown in Table R3, LatSearch exhibits predictable, monotonic improvement as the search budget grows from $N=4$ to $N=8$. This validates that the search mechanism behaves consistently and that quality improvements scale with additional compute.
>
> Temperature $\tau$ Sensitivity: Varying $\tau \in \lbrace 0.5,\, 1.0,\, 2.0 \rbrace$ yields only modest variation in VBench-2.0 performance (Table R4), indicating that the resampling mechanism is stable across a broad range.
>
> Scoring-Timestep $S$ Sensitivity: The performance trend in Table R5 reflects a natural trade-off in latent-space scoring. Because the latent reward model is trained on approximated (similarity-derived) credits rather than ground-truth labels, its predictions inevitably contain uncertainty. With too few scoring points, the search procedure lacks sufficient temporal coverage and may fail to reliably distinguish candidate trajectories. However, using too many scoring points ($\ge 4$) amplifies prediction variance and accumulates noise across timesteps, which leads to biased aggregated rewards and degraded final performance. Our results show that a moderate number of well-spaced scoring timesteps $\lbrace 10, 15, 20\rbrace$ achieves the best balance between stability and discriminative power.
>
> **Table R3. Video Generation Results with Varied Search Budgets.**
>
> |      Methods      | Creativity | Commonsense | Controllability | Human Fidelity | Physics |         Average         | Inference Time |
> | :---------------: | :--------: | :---------: | :-------------: | :------------: | :-----: | :---------------------: | :-------------: |
> |     Baseline     |   53.81   |    55.63    |      21.99      |     82.11     |  45.98  |     **51.90**     |  77.21 ± 0.26  |
> | + LatSearch (N=4) |   57.70   |    54.70    |      22.00      |     83.63     |  46.03  | **52.81 (+0.91)** | 132.71 ± 2.55 |
> | + LatSearch (N=6) |   58.12   |    59.37    |      22.70      |     82.59     |  46.44  | **53.84 (+1.94)** | 182.43 ± 6.53 |
> | + LatSearch (N=8) |   58.47   |    58.87    |      22.26      |     84.00     |  47.05  | **54.13 (+2.23)** | 225.56 ± 15.57 |
>
> **Table R4. Sensitivity analysis of the temperature parameter $\tau$ on VBench-2.0.**
>
> | Temperature | Creativity | Commonsense | Controllability | Human Fidelity | Physics |     Average     |
> | :---------: | :--------: | :---------: | :-------------: | :------------: | :-----: | :-------------: |
> |     0.5     |   56.74   |    58.25    |      21.83      |     84.53     |  46.44  | **53.56** |
> |     1.0     |   58.12   |    59.37    |      22.69      |     82.59     |  46.44  | **53.84** |
> |     2.0     |   58.42   |    58.23    |      21.56      |     82.12     |  46.32  | **53.33** |
>
> **Table R5. Sensitivity analysis of scoring timestep schedules on VBench-2.0.**
>
> |          Searching Scheduler          | Creativity | Commonsense | Controllability | Human Fidelity | Physics |     Average     |
> | :------------------------------------: | :--------: | :---------: | :-------------: | :------------: | :-----: | :-------------: |
> |       $\lbrace 10, 15\rbrace$       |   58.01   |    56.16    |      22.08      |     82.38     |  42.22  | **52.17** |
> |     $\lbrace 10, 15, 20\rbrace$     |   58.12   |    59.37    |      22.69      |     82.59     |  46.44  | **53.84** |
> |   $\lbrace 10, 15, 20, 25\rbrace$   |   57.30   |    56.80    |      22.66      |     82.71     |  46.61  | **53.22** |
> | $\lbrace 10, 15, 20, 25, 30\rbrace$ |   55.40   |    57.63    |      21.59      |     81.72     |  45.34  | **52.34** |

---

> ### Author Response · Authors · 2025-11-20
> **Official Comment by Authors [3/3]**
>
> - **Q4. No hyper-parameters optimization for EvoSearch under Wan2.1-1.3B model.**
>
> **A4.** We thank the reviewer for the concern. The EvoSearch hyper-parameters we used are not defaults, but the official settings optimized by the EvoSearch authors specifically for the Wan2.1-1.3B model, as documented in their public implementation. Using these officially optimized settings ensures the fairest comparison.
>
> - **Q5. Timing does not clarify if the latent reward model cost is included.**
>
> **A5.** We thank the reviewer for the question. The reported wall-clock times fully include the cost of the latent reward model. For transparency, we provide a detailed per-component runtime breakdown (DiT, VAE decoder, and latent reward model) in Table R6 and R7. These results confirm that the latent reward model accounts for only a small portion of overall runtime, and that the efficiency gains of LatSearch remain substantial even when all components are included.
>
> **Table R6. Runtime of each module.**
>
> |       Module       |   Inference Time (sec.)   |
> | :-----------------: | :-----------------------: |
> |     DiT Forward     |   1.35±0.13 (per step)   |
> |     VAE Decoder     |  1.85±0.26 (per video)  |
> | Latent Reward Model | 0.84±0.003 (per latent) |
>
> **Table R7. Complete runtime comparison.**
>
> |   Methods   |   DiT Time   | Decoder Time | Reward Time |  Total Time  | Vbench-2.0 Results |
> | :---------: | :----------: | :----------: | :---------: | :----------: | :-----------------: |
> |  Baseline  | 67.46±0.71 |  1.85±0.26  |      0      | 69.31±0.76 |   **51.82**   |
> | VideoReward | 266.71±0.40 | 10.12±0.54 | 0.56±0.33 | 277.39±0.75 |   **52.80**   |
> |  EvoSearch  | 756.66±1.27 | 31.99±0.96 | 1.92±1.04 | 790.57±1.90 |   **55.01**   |
> |  LatSearch  | 156.11±5.11 |  1.88±0.24  | 1.03±0.06 | 159.02±5.12 |   **55.25**   |

---

> ### Author Response · Authors · 2025-11-26
>
> Dear Reviewer ubZH,
>
> Thank you again for your constructive feedback. We have addressed all concerns, added new experiments, clarified technical details, and updated the manuscript accordingly.
>
> If any points require further clarification, we would be happy to provide additional details. Should the revisions resolve your earlier concerns, we kindly invite you to reconsider the score.
>
> Thank you for your time and helpful evaluation.
>
> Best regards,
>
> The Authors

---

### Official Review · Reviewer_Pxsh · 2025-10-26

**Soundness:** 2
**Presentation:** 2
**Contribution:** 3
**Rating:** 4
**Confidence:** 4

**Summary:**

The paper introduces a latent reward model that predicts visual quality, motion quality, and text alignment directly from intermediate diffusion latents, using similarity-grounded credit assignment to convert final video-level rewards into per-latent supervision. A transformer model trained with regression + preference losses enables test-time branching and pruning guided purely in latent space. Experiments on VBench-2.0 show a better trade-off with recent test-time scaling baselines, at lower computational cost, and ablations confirm the necessity of latent-level supervision and balanced preference learning.

**Strengths:**

- Proposing a latent-level reward model that enables test-time search without decoding videos substantially reduces overhead while preserving or improving final quality, which is particularly impactful for heavy modalities such as video diffusion. Moreover, the paper attempts to address a fundamental obstacle in test-time search for diffusion models (i.e., prohibitive decoding cost with VAEs), and this attempt is valuable.
- The method is evaluated under state-of-the-art settings, using standardized benchmarks such as VBench-2.0 and strong modern text-to-video backbones, and the results are reported in a detailed manner.

**Weaknesses:**

1. The paper does not show scaling of LatSearch with respect to the search budget. Table 1 suggests that under equal compute, LatSearch does not outperform EvoSearch. It remains unclear whether increasing the latent search budget would close this gap. Without such evidence, the claimed efficiency-quality trade-off of latent-level search remains only partially substantiated.

2. No human preference study is provided. While VBench-2.0 benchmark metrics improve, it is unclear whether the proposed search strategy actually improves perceived human preference. Without such evidence, overfitting to automated metrics cannot be ruled out.

3. Similar to weakness 2, the validity of the latent reward model is not empirically anchored to human judgments or other reliable evaluators. For example, correlating latent scores with human preferences after a normal denoising process (i.e., denoising $z_t$ over $t$ steps and then decoding) would strengthen the claim. Without such grounding, the reliability of latent supervision remains unestablished.

**Questions:**

- See weakness for major questions.
- The paper employs VideoReward as the reward model. To what extent is the proposed latent-level search compatible with existing guidance methods (e.g., Universal Guidance), and how much computational cost would they incur in practice?
- The paper reports substantial wall-clock gains (e.g., 133s vs. 783s in Table 1), but the breakdown is unclear. Could you provide a per-component runtime profile under your primary setting: DiT forward per denoising step and VAE decode per video? In particular, how large is the VAE decoding cost relative to DiT compute, and how does this explain LatSearch vs. EvoSearch/VideoReward?

---

> ### Author Response · Authors · 2025-11-20
> **Official Comment by Authors [1/2]**
>
> Thank you for recognizing the motivation and significance of latent-level search and for your positive remarks on our experimental setup. We appreciate your constructive feedback. We provide detailed responses to each point below.
>
> - **Q1. The paper does not show scaling of LatSearch with respect to the search budget. Table 1 suggests that under equal compute, LatSearch does not outperform EvoSearch.**
>
> **A1.** Thanks for the helpful question. To clarify, LatSearch does outperform EvoSearch when evaluated under exactly the same setting. As shown in Table 1 of the manuscript, when both methods use the same sampler, LatSearch achieves a higher VBench-2.0 score (55.25 vs. 55.01) while requiring 4.8 times less inference time. For completeness, we also provide additional results reporting the GPU memory usage of both EvoSearch and LatSearch during inference (Table R1), confirming that the comparison is fair and measured under identical conditions.
>
> To further address the reviewer’s question regarding scaling behavior, we include a search-budget ablation (N = 4/6/8), which shows clear and monotonic quality improvements as compute increases (Table R2).
>
> **Table R1. Comparison of GPU Memory, Inference Time, and VBench-2.0 Results.**
>
> |  Methods  | GPU Memory | Inference Time | VBench2.0 Results |
> | :-------: | :--------: | :------------: | :---------------: |
> | EvoSearch |  19.88 GB  | 783.76 ± 3.15 |       55.01       |
> | LatSearch |  20.36 GB  | 164.41 ± 4.79 |       55.25       |
>
> **Table R2. Video Generation Results with Varied Search Budgets.**
>
> |      Methods      | Creativity | Commonsense | Controllability | Human Fidelity | Physics |         Average         |
> | :---------------: | :--------: | :---------: | :-------------: | :------------: | :-----: | :---------------------: |
> |     Baseline     |   53.81   |    55.63    |      21.99      |     82.11     |  45.98  |     **51.90**     |
> | + LatSearch (N=4) |   57.70   |    54.70    |      22.00      |     83.63     |  46.03  | **52.81 (+0.91)** |
> | + LatSearch (N=6) |   58.12   |    59.37    |      22.70      |     82.59     |  46.44  | **53.84 (+1.94)** |
> | + LatSearch (N=8) |   58.47   |    58.87    |      22.26      |     84.00     |  47.05  | **54.13 (+2.23)** |
>
> - **Q2. No human preference study is provided.**
>
> **A2.** Thanks for the suggestion. While VBench-2.0 is designed to closely match human judgments and its benchmark paper reports strong agreement with human preference trends, we agree that direct human evaluation is valuable. We are currently collecting additional video preference from more participants, once the study is complete, we will update results here, thanks!
>
> - **Q3. Correlating latent scores with human preference (after decoding) would help.**
>
> **A3.** Thank you for the helpful suggestion. We agree that directly aligning latent scores with human preference is valuable, but we note that our goal is to improve final video generation, and a good intermediate latent does not always correspond to a good final video because quality emerges from the entire denoising trajectory.
>
> Nevertheless, we provide evidence that latent rewards are meaningfully aligned with final video quality. As shown in Table 5 in the manuscript, LRM predictions correlate with video-level performance, and all credit-assignment strategies produce reward models that consistently improve downstream video generation. In addition, our results in downstream video generation show that LatSearch improves perceived video quality across all criteria. These results indicate that the latent reward model provides a reliable supervision signal for guiding test-time search.

---

> > ### Author Response · Authors · 2025-11-25
> > **Human Preference Study Results**
> >
> > We have now completed our human preference study. Specifically, we sampled 40 video pairs generated by the baseline model and LatSearch across diverse categories (animals, humans, natural scenery, and architecture), and asked 30 participants to choose the better video in each pair. Preferences were evaluated along three dimensions: Visual Quality (clarity, details, absence of artifacts), Motion Quality (temporal consistency, smoothness, realism), and Video–Text Alignment (subject/action/scene fidelity to the prompt). Participants consistently preferred LatSearch across all dimensions. The results have been added to the revised manuscript.
> >
> > **Table R5. Human pairwise preference rates.**
> >
> > |     Criterion     | Preference for LatSearch |
> > | :---------------: | :----------------------: |
> > |  Visual Quality   |         72.15%           |
> > |  Motion Quality   |         75.29%           |
> > |  Text Alignment   |         78.51%           |
> > |    **Average**    |       **75.32%**         |

---

> > > ### Comment · Reviewer_Pxsh · 2025-11-26
> > >
> > > Thank you for your elaboration. Now that my initial concerns are partially resolved, I would like you to ask additional questions.
> > >
> > > AQ1: In A4, you mentioned your lightweight evaluation as follows.
> > >
> > > > The latent reward model is called only at selected timesteps (e.g., 10/15/20)
> > >
> > > - Why did you choose these specific timesteps?
> > > - And can you show results when using other timesteps? For example, I am interested in comparisons where search is applied at all timesteps, as well as results obtained by varying the selected timesteps in different ways.
> > > - Why don't you apply your search for timestep=5 in Table 7?
> > >
> > > I would like to see more detailed discussion about the relationship between selected timesteps and performance gain.

---

> ### Author Response · Authors · 2025-11-20
> **Official Comment by Authors [2/2]**
>
> - **Q4. To what extent is LatSearch compatible with existing guidance methods (e.g., Universal Guidance)? What cost would they incur?**
>
> **A4.** Thank you for the insightful question. Our latent reward model (LRM) is fully differentiable, and thus LatSearch is in principle compatible with Universal Guidance–style gradient-based updates. Specifically, one could directly compute $\frac{\partial R}{\partial z_t}$ from our LRM and perform gradient ascent on the latent to maximize the reward, exactly as in classifier or reward guidance.
>
> However, such a gradient-based integration would introduce substantial computational overhead. Although Our LRM's forward inference is lightweight, backpropagating through it at every denoising step (and often for multiple gradient iterations per step) would be extremely costly in both memory and runtime. This would largely negate the efficiency benefits of inference-time search. By contrast, LatSearch is deliberately designed to be fast and decoding-free, using only forward passes of the LRM at a small number of sparse timesteps. This avoids all gradient computation and makes our method faster than inference-time optimization baselines.
>
> - **Q5. Runtime breakdown and explanation of LatSearch efficiency.**
>
> **A5.** Thank you for the insightful question. To make the source of the reported wall-clock improvements clear, we provide a detailed per-module runtime breakdown under our primary experimental setting, measured on an A100 (80GB). The three major components of inference are: (1) DiT forward passes during denoising, (2) VAE decoding of the final video, and (3) the latent reward model (LRM) evaluation.
>
> As shown in Table R3, the results indicate that VAE decoding is more expensive than a single DiT step, and repeated decoding, as required by VideoReward and EvoSearch, quickly becomes the dominant inference cost. In contrast, LatSearch avoids decoding entirely during search and evaluates the reward model only at a small number of sparse timesteps, keeping the overhead minimal.
>
> Furthermore, results in Table R4 present that LatSearch achieves 4–5 times faster wall-clock time than EvoSearch while achieving higher VBench-2.0 scores under identical sampler settings. This efficiency advantage comes from two design choices: (1) Decoding-free search. Unlike VideoReward and EvoSearch, LatSearch never decodes intermediate videos, eliminating the most expensive component of inference. (2) Sparse and lightweight reward evaluation. The latent reward model is called only at selected timesteps (e.g., 10/15/20), rather than at every diffusion step.
>
> Overall, the runtime analysis confirms that the efficiency gains of LatSearch come from a fundamental algorithmic shift, moving reward evaluation to latent space and eliminating repeated decoding, rather than implementation or engineering details. We sincerely thank the reviewer for requesting this breakdown, as it allows us to more clearly highlight the core contribution and efficiency advantages of our method.
>
> **Table R3. Runtime of each module.**
>
> |       Module       |   Inference Time (sec.)   |
> | :-----------------: | :-----------------------: |
> |     DiT Forward     |   1.35±0.13 (per step)   |
> |     VAE Decoder     |  1.85±0.26 (per video)  |
> | Latent Reward Model | 0.84±0.003 (per latent) |
>
> **Table R4. Complete runtime comparison.**
>
> |   Methods   |   DiT Time   | Decoder Time | Reward Time |  Total Time  | Vbench-2.0 Results |
> | :---------: | :----------: | :----------: | :---------: | :----------: | :-----------------: |
> |  Baseline  | 67.46±0.71 |  1.85±0.26  |      0      | 69.31±0.76 |   **51.82**   |
> | VideoReward | 266.71±0.40 | 10.12±0.54 | 0.56±0.33 | 277.39±0.75 |   **52.80**   |
> |  EvoSearch  | 756.66±1.27 | 31.99±0.96 | 1.92±1.04 | 790.57±1.90 |   **55.01**   |
> |  LatSearch  | 156.11±5.11 |  1.88±0.24  | 1.03±0.06 | 159.02±5.12 |   **55.25**   |

---

> ### Author Response · Authors · 2025-11-26
>
> Thank you for the helpful suggestion.
>
> - **Why scoring is not applied at very early timesteps and not applied at all timesteps**
>
> We avoid applying search at very early timesteps because latents in these stages are dominated by noise and contain almost no semantic information. Reward evaluation is therefore highly unstable, and scores become non-informative for candidate selection.
>
> On the other hand, we do not apply scoring at very late timesteps primarily due to computational cost. For example, with 6 candidates and a DiT forward time of 1.35s per step, scoring within the [10,20] range increases the total inference cost by approximately 67.5–135 seconds (best–worst case). However, scoring within [30,40] would increase the cost to 202–270 seconds, resulting in a 2–3 times higher runtime. This conflicts with our key motivation:
>
> > A major bottleneck in inference-time scaling for video diffusion is the inability to evaluate intermediate latents reliably and early. Without early latent-level assessments, the model cannot stop search early for efficiency, and errors introduced at initial stages propagate through the entire long denoising chain.
>
> Therefore, we intentionally apply search as early as possible after semantically meaningful structure emerges, achieving a favorable quality–efficiency balance.
>
> Furthermore, our experiments show that adding additional late scoring steps does not improve results (Table R6). This is because the latent reward model inevitably carries some estimation uncertainty, and increasing the number of scoring points accumulates this uncertainty, eventually degrading search quality.
>
> **Table R6. Sensitivity analysis of scoring timestep schedules on VBench-2.0.**
>
> |          Searching Scheduler         | Creativity | Commonsense | Controllability | Human Fidelity | Physics |     Average       | Inference Time  |
> | :----------------------------------: | :--------: | :---------: | :-------------: | :------------: | :-----: | :-------------:   | :-------------: |
> |                 Baseline             |   53.81    |    55.63    |      21.99      |     82.11      |  45.98  |     **51.90**     |  77.21 ± 0.26   |
> |       $\lbrace 10, 15\rbrace$        |   58.01    |    56.16    |      22.08      |     82.38      |  42.22  | **52.17 (+0.27)** |  154.63 ± 3.06  |
> |     $\lbrace 10, 15, 20\rbrace$      |   58.12    |    59.37    |      22.69      |     82.59      |  46.44  | **53.84 (+1.94)** |  182.43 ± 6.53  |
> |   $\lbrace 10, 15, 20, 25\rbrace$    |   57.30    |    56.80    |      22.66      |     82.71      |  46.61  | **53.22 (+1.32)** |  186.47 ± 8.58  |
> | $\lbrace 10, 15, 20, 25, 30\rbrace$  |   55.40    |    57.63    |      21.59      |     81.72      |  45.34  | **52.34 (+0.44)** |  198.48 ± 10.99 |
>
> **We thank the reviewer for the constructive comments and hope our clarifications resolve the concerns raised. We have incorporated these discussions into the revised manuscript (Appendix A.4.4), and we sincerely appreciate the reviewer’s feedback for helping us improve the soundness of our work.**

---

> > ### Comment · Reviewer_Pxsh · 2025-11-27
> >
> > Thank you for your prompt response again. I now understand the rationale behind applying your search method to selected timesteps.
> >
> > However, this has raised a new concern about the validity of your LatSearch evaluator.
> > > This is because the latent reward model inevitably carries some estimation uncertainty, and increasing the number of scoring points accumulates this uncertainty, eventually degrading search quality.
> >
> > I would like to ask additional questions to clarify whether your method is sound.
> >
> > AQ2. Could you explain why the Diversity of LatSearch is lower than that of previous methods?
> >
> > AQ3. I have some doubts about the accuracy of the reward estimation in your LatSearch evaluator. To address this, could you please calculate the Mean Squared Error (MSE) at timesteps $t = \{0, 5, 10, 15, 20, 25, \dots\}$ for the following two values against the ground truth?
> >
> > 1. The score predicted by the LatSearch evaluator for $z_t$.
> >
> > 2. The score obtained by decoding $z_t$ via VAE and evaluating it with the original evaluator.
> >
> > For the ground truth, please use the score obtained by completing the generation process from $z_t$ to $z_0$, decoding the final $z_0$ via VAE, and evaluating it with the original evaluator.

---

> > > ### Author Response · Authors · 2025-12-01
> > >
> > > - **AQ2. Why the Diversity of LatSearch is lower than that of previous methods.**
> > >
> > > This claim is not true. We clarify that LatSearch actually improves Diversity over the baseline and most prior methods. Specifically, LatSearch increases Diversity from 65.85 to 69.19, outperforming FreeInit, FreqPrior, and VideoReward. While EvoSearch reports a slightly higher score, LatSearch achieves competitive diversity with 4.8 times faster inference under the same sampling configuration.
> > >
> > > Moreover, LatSearch is designed primarily to enhance video quality and inference efficiency, rather than explicitly maximising stochastic diversity. The method focuses on identifying high-reward trajectories efficiently, which naturally results in a more measured level of variation compared to mutation-heavy strategies.
> > >
> > > - **AQ3. MSE comparison between latent-reward vs. ground-truth and decoded-reward vs. ground-truth.**
> > >
> > > Thank you for raising this concern. We have conducted the requested experiment and report the MSE between (1) the latent-reward predictions and (2) decoded-reward predictions, each compared against the final video reward at multiple timesteps.
> > >
> > > The results show a clear and consistent gap: latent-reward predictions have 3–5 times lower MSE than decoded-reward predictions at every timestep, demonstrating that the latent reward model provides a substantially more accurate estimate of the eventual video quality.
> > >
> > > This directly addresses the reviewer’s concern regarding whether latent-space rewards can reliably approximate final outcomes. In fact, the experiment confirms that:
> > >
> > >  • Latent-based scores align much more closely with the final video reward,
> > >
> > >  • Decoded intermediate rewards are significantly noisier, due to decoding artifacts and inconsistencies in partially denoised frames,
> > >
> > >  • LRM provides a more stable supervision signal, strengthening its role in inference-time search.
> > >
> > > Therefore, this experiment not only resolves the reviewer’s concern but also highlights the core contribution of LatSearch: high-fidelity latent evaluation enables early, decoding-free trajectory selection, which would not be feasible using decoded-frame rewards.
> > >
> > > We have added this analysis and more detailed results to the revised manuscript (Appendix A.4.8).
> > >
> > > **Table R7. MSE between latent-reward/decoded-reward scores and the final video reward across timesteps.**
> > >
> > > |   **Methods**   | **t=0** | **t=5** | **t=10** | **t=15** | **t=20** | **t=25** | **t=30** | **t=35** | **t=40** |
> > > | :-------------------: | ------------- | ------------- | -------------- | -------------- | -------------- | -------------- | -------------- | -------------- | :------------: |
> > > | **Decoded MSE** | 55.52         | 55.41         | 54.97          | 54.17          | 53.44          | 52.87          | 52.27          | 50.79          |     43.45     |
> > > | **Latent MSE** | 15.66         | 14.90         | 14.00          | 13.20          | 12.56          | 11.90          | 11.40          | 10.92          |     10.50     |

---

### Official Review · Reviewer_Z2L1 · 2025-10-30

**Soundness:** 3
**Presentation:** 3
**Contribution:** 3
**Rating:** 6
**Confidence:** 4

**Summary:**

This paper proposes LatSearch, an efficient inference-time optimization method for video diffusion models. It introduces a latent reward model that evaluates partially denoised latents at any timestep, providing real-time feedback on visual quality, motion quality, and text alignment. Based on this, the authors design a Reward-Guided Resampling and Pruning (RGRP) algorithm that dynamically retains high-quality candidates and discards low-quality ones during generation, reducing unnecessary computation. Experiments on the VBench 2.0 benchmark show that LatSearch achieves comparable or better video quality while reducing inference time by up to 79%. Overall, this approach enables more efficient and controllable video generation through latent-space evaluation and optimization.

**Strengths:**

1. This work proposes a novel latent-space reward model that can directly evaluate partially denoised latents, offering fine-grained supervision during generation and turning sparse, delayed feedback into meaningful intermediate guidance.

2. This work introduces Reward-Guided Resampling and Pruning, which uses probabilistic resampling and cumulative reward selection to retain promising trajectories and remove weak ones, effectively improving both quality and efficiency.

3. This work presents strong empirical results on the VBench 2.0 benchmark, showing consistent gains across multiple metrics and achieving up to 79% faster inference compared with existing methods, demonstrating solid practical value.

**Weaknesses:**

1. Lack of theoretical grounding on the convergence or optimality of the proposed resampling and pruning process, making it unclear whether the cumulative reward aggregation in latent space guarantees consistent improvement across diffusion steps.

2. Lack of deeper analysis on the similarity-based credit assignment used to construct latent-level rewards, as the cosine similarity between intermediate and final latents may not reliably capture semantic contribution at different timesteps, potentially introducing biased supervision.

3. Lack of discussion on the generalization of the latent reward model to other video diffusion backbones or reward verifiers, since all training and evaluation are conducted within the same model family (Wan2.1 and Qwen2-VL), leaving cross-model robustness untested.

4. Lack of an ablation comparing different latent similarity metrics (e.g., learned similarity networks vs cosine similarity) to validate the effectiveness of the similarity-grounded credit assignment.

**Questions:**

1. How sensitive is the performance of LatSearch to the specific design of the latent reward model? For example, would replacing the cosine-based similarity grounding with a learned temporal weighting or a contrastive objective further improve reward accuracy and stability?

2. Could the authors clarify whether the reward model or the RGRP strategy can be extended to cross-modal tasks, such as audio–video generation or video editing? It would be interesting to understand how latent-space rewards generalize beyond text-to-video synthesis.

---

> ### Author Response · Authors · 2025-11-20
> **Official Comment by Authors [1/2]**
>
> Thank you for your encouraging review and your positive assessment of our method’s design, effectiveness, and efficiency. We are grateful for your insightful feedback. We respond to each of your comments in detail below.
>
> - **Q1. Lack of theoretical grounding on the convergence or optimality of RGRP in latent space.**
>
> **A1.** We thank the reviewer for the insightful comment. We clarify that our objective is to design an inference-time candidate search mechanism for diffusion models under limited compute. Because both the diffusion dynamics and the latent reward model are non-convex and learned, we do not claim global optimality or formal convergence guarantees. We agree that establishing convergence for such a procedure is inherently non-trivial. While our resampling–pruning procedure is conceptually inspired by Sequential Monte Carlo (SMC) methods, and prior work on adaptive SMC methods provides theoretical convergence analyses, including WLLN and CLT results for adaptive resampling and proposal adaptation (Beskos et al., 2014), these analyses rely on exact likelihood-based importance weights and analytically defined target distributions. In contrast, our latent reward model produces *learned* and *approximate* credit estimates for intermediate denoising states, which do not satisfy the assumptions required by existing SMC theory. Therefore, providing formal convergence guarantees is outside the scope of this work, and we will explicitly clarify this limitation in the revised paper.
>
> [1]. Beskos A, Jasra A, Kantas N, Thiery A. On the Convergence of Adaptive Sequential Monte Carlo Methods. The Annals of Applied Probability. 2014.
>
> - **Q2. Lack of deeper analysis on the similarity-based credit assignment.**
>
> **A2.** We appreciate the reviewer’s suggestion and have added a more thorough analysis of our similarity-based credit assignment. Cosine similarity was originally chosen because it reflects the directional progression of latent refinement during denoising and is invariant to latent magnitude, making it well-suited for semantic credit allocation. Following the reviewer’s recommendation, we conducted new ablations comparing cosine similarity with alternative strategies, including time-based weighting and error–based weighting. These results in Tables R1 and R2 show that cosine similarity consistently provides the most stable latent supervision, yields higher reward-prediction accuracy, and produces better downstream video quality.
>
> **Table R1. Comparison of VQ, MQ, and TA accuracy across different credits assignment strategies.**
>
> |      Methods      | VQ Accuracy | MQ Accuracy | TA Accuracy |     Average     |
> | :----------------: | :---------: | :---------: | :---------: | :-------------: |
> |      Uniform      |    73.61    |    68.73    |    81.30    | **74.54** |
> |    Exponential    |    73.39    |    69.29    |    81.00    | **74.56** |
> |      L2 Error      |    75.68    |    70.23    |    81.19    | **75.70** |
> | Cosine Similarity |    75.67    |    72.01    |    80.97    | **76.22** |
>
> **Table R2. Generated video quality under different credit assignment strategies.**
>
> |      Methods      | Creativity | Commonsense | Controllability | Human Fidelity | Physics |     Average     |
> | :----------------: | :--------: | :---------: | :-------------: | :------------: | :-----: | :-------------: |
> |      Baseline      |   53.81   |    55.63    |      21.99      |     82.11     |  45.98  | **51.90** |
> |      Uniform      |   54.98   |    52.40    |      21.44      |     84.98     |  47.75  | **52.31** |
> |    Exponential    |   54.50   |    52.09    |      21.63      |     83.03     |  49.30  | **52.11** |
> |      L2 Error      |   55.78   |    56.64    |      22.20      |     82.12     |  46.95  | **52.74** |
> | Cosine Similarity |   58.12   |    59.37    |      22.69      |     82.59     |  46.44  | **53.84** |
>
> - **Q3. No ablation comparing different latent similarity metrics (e.g., learned similarity network).**
>
> **A3.** We agree that learning a temporal similarity function is an excellent idea and could in principle yield more precise credit assignment. However, training such a network would require manually annotating the semantic contribution of each intermediate latent, which is extremely expensive and beyond the scope of our problem setting. Our focus is not on learning a new similarity estimator, but on how to train the latent reward model with an approximated credit assignment.
>
> Importantly, our experiments show that cosine-based similarity already provides a strong and stable supervision signal for training the latent reward model. To further address the reviewer’s concern, and as detailed in our response to Q2, we additionally compared several alternative non-learned strategies. Across all metrics, cosine assignment consistently yields the highest reward-model accuracy and the strongest downstream video quality, demonstrating that our method is robust to the choice of similarity measure.

---

> ### Author Response · Authors · 2025-11-20
> **Official Comment by Authors [2/2]**
>
> - **Q4. Lack of discussion on generalization to other backbones or verifiers.**
>
> **A4.** We thank the reviewer for raising this important point. We have addressed this concern by conducting additional experiments on other backbones. LatSearch is designed to be backbone-agnostic because it operates purely in latent space and does not rely on any model-specific architectural components. To demonstrate this generality, we additionally evaluate LatSearch on a much larger backbone (Wan2.1-14B). As shown in Table R5, LatSearch consistently improves video quality across all dimensions, confirming that the method generalizes beyond the 1.3B base model.
>
> Regarding reward verifiers, we agree that evaluating multiple verifiers is valuable. However, training a new video-reward model is a substantial undertaking that requires large-scale multimodal data and extensive computational resources. We therefore adopt Qwen2-VL-3B, which is currently one of the strongest publicly available video-language models and has demonstrated robust performance across a wide range of video understanding tasks. Since LatSearch interacts with the verifier only through scalar reward signals, and does not rely on any specific architectural assumptions, we expect it to remain compatible with alternative verifiers.
>
> **Table R5. Video Generation Results on the Wan2.1-14B Backbone.**
>
> |   Methods   | Creativity | Commonsense | Controllability | Human Fidelity | Physics |         Average         |
> | :---------: | :--------: | :---------: | :-------------: | :------------: | :-----: | :---------------------: |
> |  Baseline  |   55.21   |    56.18    |      21.79      |     89.74     |  39.96  |     **52.58**     |
> | + LatSearch |   56.28   |    57.64    |      22.52      |     91.45     |  40.17  | **53.61 (+1.03)** |
>
> - **Q5. Can LRM extend to audio–video or video editing tasks?**
>
> **A5.** We appreciate the reviewer for highlighting this meaningful extension. In principle, the latent reward model is not restricted to text-to-video generation: two of its core evaluation dimensions—visual quality and motion quality—are modality-agnostic and directly applicable to any video synthesis or video editing pipeline. Our framework provides a general latent-level evaluation and trajectory selection mechanism, which can be readily adopted as long as the target task exposes partially denoised latents.
>
> For audio–video or editing scenarios, the third dimension (text alignment) can naturally be replaced with other task-specific alignment signals (e.g., audio–motion synchronization or edit consistency). This makes LatSearch flexible and adaptable beyond the T2V setting. We will make our best efforts to extend our latent-space reward guidance to video editing as part of our ongoing future work.

---

> ### Author Response · Authors · 2025-11-26
>
> Dear Reviewer Z2L1,
>
> Thank you again for your constructive feedback. We have addressed all concerns, added new experiments, clarified technical details, and updated the manuscript accordingly.
>
> If any points require further clarification, we would be happy to provide additional details.
> Should the revisions resolve your earlier concerns, we kindly invite you to reconsider the score.
>
> Thank you for your time and helpful evaluation.
>
>
> Best regards,
>
> The Authors

---

### Official Review · Reviewer_uF4N · 2025-11-02

**Soundness:** 2
**Presentation:** 3
**Contribution:** 3
**Rating:** 4
**Confidence:** 3

**Summary:**

This paper proposes LATSEARCH, a plug-in inference-time search framework for video diffusion that trains a latent reward model to score intermediate (partially denoised) representations on visual quality, motion quality, and text alignment, and then applies reward-guided resampling & pruning to guide trajectory search in diffusion models using this reward model. Experiments show that, compared to state-of-the-art video verification methods, it achieves higher final video quality while suppressing computational overhead, operates significantly faster, and delivers performance equal to or better than strong inference baselines.

**Strengths:**

- The paper is well-organized and easy to follow.
- The authors propose reducing execution time in video diffusion by estimating rewards in intermediate latent variables and performing exploration based on them, which is both practical and novel.
- Demonstrates competitive or better quality than strong inference-time baselines while reducing runtime, showing real deployment potential.

**Weaknesses:**

- In this work, when constructing the latent reward dataset, the authors compute the cosine similarity between the current latent variable and the final latent variable, and weight the reward by this similarity to adjust the proportion of reward assigned at intermediate timesteps. However, this choice is largely intuitive, and other credit-assignment strategies are conceivable. For example, one could set s to be the error between the current and final latents (e.g., squared error), or simply apply time-based weighting of the reward. The authors should explain why their choice is preferable to such alternatives and provide lightweight ablations comparing (i) time-based schedules (e.g., uniform, exponential), and (ii) error- or SNR-based weighting (e.g., L2 error or denoiser loss).
- The proposed method clearly depends on the performance of the Latent Reward Model (LRM). While the paper shows improvements when using the LRM within the search procedure, the LRM itself is not evaluated in isolation. Without such an evaluation, it is difficult to assess the validity of the Latent Reward Data Construction, i.e., whether the problem is well-posed and learnable. In other words, (A) whether latent rewards can be accurately predicted and (B) whether, given accurate latent rewards, the search can effectively leverage them are two distinct questions that should be validated separately.

**Questions:**

Regarding the weaknesses noted above, please respond and, where appropriate, include additional experiments to address them. Also, the term Similarity-Grounded Credit Assignment (SGCA) appears only once in the paper, so defining it as a term is not justified. Please either use the term consistently throughout the paper or remove it.

---

> ### Author Response · Authors · 2025-11-20
> **Official Comment by Authors [1/2]**
>
> Thank you for your positive remarks on the paper’s clarity, novelty, and practical value. We appreciate your thoughtful evaluation. We address all your questions and concerns point-by-point below.
>
> - **Q1. Why choose cosine similarity, provide a ablation comparison to other weighting methods (time-based and error-based schedules).**
>
> **A1.** Thank you for the question and the valuable suggestions. We chose cosine similarity because it naturally captures the directional progression of the latent trajectory during denoising, which aligns closely with semantic refinement in diffusion models. While L2 distance is also a valid choice, it is more sensitive to latent magnitude variations across timesteps, whereas cosine provides scale-invariant, structure-oriented credit that better reflects latent alignment. Time-based schedules (uniform or exponential) do not account for latent evolution and may assign credit unrelated to semantic contribution.
>
> As suggested, we added ablations using L2-based, uniform, and exponential weighting. Across all settings, cosine-based credit assignment provides the most stable latent supervision, leading to improved reward-model accuracy and superior video generation quality. Although the reward-prediction accuracies in Table R1 differ only modestly, cosine supervision generalizes better to unseen latents during test-time search, resulting in consistently stronger downstream video quality (Table R2).
>
> **Table R1. Comparison of VQ, MQ, and TA accuracy across different credits assignment strategies.**
>
> |      Methods      | VQ Accuracy | MQ Accuracy | TA Accuracy |     Average     |
> | :----------------: | :---------: | :---------: | :---------: | :-------------: |
> |      Uniform      |    73.61    |    68.73    |    81.30    | **74.54** |
> |    Exponential    |    73.39    |    69.29    |    81.00    | **74.56** |
> |      L2 Error      |    75.68    |    70.23    |    81.19    | **75.70** |
> | Cosine Similarity |    75.67    |    72.01    |    80.97    | **76.22** |
>
> **Table R2. Generated video quality under different credit assignment strategies.**
>
> |      Methods      | Creativity | Commonsense | Controllability | Human Fidelity | Physics |     Average     |
> | :----------------: | :--------: | :---------: | :-------------: | :------------: | :-----: | :-------------: |
> |      Baseline      |   53.81   |    55.63    |      21.99      |     82.11     |  45.98  | **51.90** |
> |      Uniform      |   54.98   |    52.40    |      21.44      |     84.98     |  47.75  | **52.31** |
> |    Exponential    |   54.50   |    52.09    |      21.63      |     83.03     |  49.30  | **52.11** |
> |      L2 Error      |   55.78   |    56.64    |      22.20      |     82.12     |  46.95  | **52.74** |
> | Cosine Similarity |   58.12   |    59.37    |      22.69      |     82.59     |  46.44  | **53.84** |
>
> - **Q2. The LRM itself is not evaluated in isolation, and whether latent rewards can be accurately predicted.**
>
> **A2.** We thank the reviewer for highlighting this important point. We note that the latent reward model (LRM) is evaluated in isolation in our manuscript (Figure 4 and Table 5), and we also present the relevant results below for the reviewer’s convenience. These analyses demonstrate that latent rewards are learnable and can be predicted with good reliability.
>
> First, incorporating a preference loss significantly improves the LRM’s discriminative accuracy, yielding a +3.09 average gain across VQ/MQ/TA (Table R3), indicating that intermediate latents contain sufficient semantic information for accurate reward prediction. Second, as shown in Table 5 of the manuscript, the model exhibits strong cross-timestep consistency, suggesting that the latent-to-quality mapping is stable and well-posed. Finally, the consistent improvements in downstream video generation quality (Table R4), across multiple credit-assignment strategies, further confirm that the predicted latent rewards are accurate enough to guide effective trajectory selection during inference.
>
> **Table R3. Comparing latent reward accuracy with and without preference loss. PL denotes preference loss.**
>
> |     Methods     |  VQ Accuracy  |  MQ Accuracy  |  TA Accuracy  |         Average         |
> | :-------------: | :-----------: | :-----------: | :-----------: | :---------------------: |
> | Reward (w/o PL) |     73.00     |     68.33     |     78.05     |     **73.13**     |
> | Reward (w/ PL) | 75.67 (+2.67) | 72.01 (+3.68) | 80.97 (+2.92) | **76.22 (+3.09)** |

---

> ### Author Response · Authors · 2025-11-20
> **Official Comment by Authors [2/2]**
>
> **Table R4. Effect of Latent Reward Model Quality on Video Generation Performance.**
>
> |      Methods      |  Creativity  |  Commonsense  | Controllability | Human Fidelity |    Physics    |         Average         |
> | :----------------: | :-----------: | :-----------: | :-------------: | :------------: | :-----------: | :---------------------: |
> |      Baseline      |     53.81     |     55.63     |      21.99      |     82.11     |     45.98     |     **51.90**     |
> | LatSearch (w/o RL) | 56.44 (+2.64) | 58.51 (+2.89) |  22.28 (+0.29)  | 84.33 (+2.22) | 45.54 (-0.45) | **53.42 (+1.52)** |
> | LatSearch (w/ PL) | 58.12 (+4.31) | 59.37 (+3.74) |  22.69 (+0.70)  | 82.59 (+0.48) | 46.44 (+0.46) | **53.84 (+1.94)** |
>
> - **Q3. Does better LRM get better search performance?**
>
> **A3.** Yes. The quantitative results consistently show that improving the LRM leads to better search performance. As shown in Table R3, adding preference loss substantially improves LRM accuracy across VQ, MQ, and TA (+3.09 on average). This translates directly into stronger test-time trajectory selection: Table R4 demonstrates that LatSearch with the improved LRM (“w/ PL”) achieves the highest video quality across all major dimensions and outperforms both the baseline model and LatSearch using the weaker LRM (“w/o PL”). This confirms that enhanced latent rewards enable more reliable resampling and pruning decisions, resulting in better final video generation.
>
> - **Q4. SGCA terminology usage.**
>
> **A4.** Thank you for pointing this out. We have removed the terminology in our revised manuscript.

---

> ### Author Response · Authors · 2025-11-26
>
> Dear Reviewer uF4N,
>
> Thank you again for your constructive feedback. We have addressed all concerns, added new experiments, clarified technical details, and updated the manuscript accordingly.
>
> If any points require further clarification, we would be happy to provide additional details.
> Should the revisions resolve your earlier concerns, we kindly invite you to reconsider the score.
>
> Thank you for your time and helpful evaluation.
>
>
> Best regards,
>
> The Authors

---

### Author Response · Authors · 2025-11-22

We thank all reviewers for their thoughtful and constructive feedback. We have updated the manuscript accordingly, and all modifications are highlighted in blue. The major changes are summarised below:
1. Removed the SGCA terminology for clarity;
2. Added experimental results on varied search budgets N (Table 2);
3. Added video generation results on the Wan2.1-14B backbone (Table 3);
4. Clarified limitations and outlined future work (Section 6);
5. Added experiments on the sensitivity of key hyperparameters (Appendix A.4.4);
6. Added experiments comparing different credit-assignment strategies (Appendix A.4.5);
7. Added an inference-time breakdown for efficiency analysis (Appendix A.4.6).

---

### Author Response · Authors · 2025-12-03
**Summary for Area Chair**

Dear Area Chair,

First of all, thank you very much for your time overseeing our submission. We fully understand that the unexpected data leakage created a substantial additional workload for area chairs, and we sincerely appreciate your effort and dedication throughout this process.

We want to provide a concise summary to support your final assessment.

Our paper received four thoughtful and constructive reviews. We are encouraged by the reviewers’ consistent recognition of our main contributions:

---

## **Key Strengths Recognised by Reviewers**

1. **Novel Latent-Space Reward Model (Z2L1, Pxsh, ubZH)**

A reward model that evaluates partially denoised latents at arbitrary timesteps, enabling dense intermediate guidance and addressing a core limitation in diffusion search.

2. **Novel Reward-Guided Resampling & Pruning (Z2L1, ubZH)**

A principled resampling and pruning mechanism that improves both quality and efficiency by focusing computation on promising trajectories.

3. **Significant Efficiency Gains Without Quality Loss (uF4N, Z2L1, ubZH)**

Up to 79% faster inference while matching or surpassing state-of-the-art video quality.

4. **Strong Practical Value and Deployment Potential (uF4N, Pxsh)**

Operating entirely in latent space eliminates expensive decoding, making test-time search practical for video diffusion.

5. **Comprehensive Experiments and Clear Presentation (uF4N, Z2L1, Pxsh)**

The paper is well-organised, easy to follow, and validated through extensive experiments on SOTA backbones and standardised benchmarks.

---

## **Concerns and How They Were Addressed**

The reviewers raised several important questions about credit assignment, hyperparameter sensitivity, latency measurements, LRM validity, and generalisation.

We took these concerns very seriously and conducted substantial new experiments during the rebuttal. Below is a concise summary of each concern and how we addressed it.

---

1. **Credit Assignment Justification**

**Concern:** Why cosine similarity over L2/uniform/exponential?
**Resolution:** Comprehensive ablations show that cosine achieves
- highest LRM accuracy (Tab. 9),
- best video quality (Tab. 10),
- strongest theoretical grounding (scale-invariance, aligned with latent semantic refinement).

**Conclusion:** Fully addressed.

---

2. **Latent Reward Model (LRM) Validity**

**Concern:** Is LRM reliable?
**Resolution:**
- Isolated LRM evaluation + preference-loss ablation (Fig. 4; Tab. 5),
- MSE analysis: latent prediction is 3–5 times more accurate than decoded-frame prediction (Fig. 6),
- Human preference study (Tab. 13).

**Conclusion:** LRM is strongly validated.

---

3. **Hyperparameter Sensitivity ($N$, $\tau$, Timesteps)**

**Concern:** Why $N=4/6$, $\tau=1.0$, timesteps $\lbrace10,15,20\rbrace$?
**Resolution:**
- Scaling with $N=4/6/8$ shows monotonic improvements (Tab. 2),
- $\tau$ robustness within $\lbrace 0.5,1.0,2.0\rbrace$ (Tab. 6),
- Timestep ablations: $\lbrace10,15,20\rbrace$ yields the best trade-off, while early or dense scoring is noisy (Tab. 8),
- Additional explanation in Sec. A.4.4.

**Conclusion:** Design choices are empirically justified.

---

4. **Runtime Breakdown**

**Concern:** Does runtime include LRM cost? Why faster?
**Resolution:**
- Full decomposition provided (Tab. 11–12),
- LRM cost is small (≈0.84 s per latent) and included,
- Baselines dominated by VAE decoding; LatSearch avoids decoding, which achieves 4–5 times speedup.

**Conclusion:** Runtime measurements are complete and transparent.

---

5. **Comparison Fairness & Generalisation**

**Concern:** EvoSearch settings and backbone generalisation.
**Resolution:**
- EvoSearch uses its official optimised setup (Sec. A.3),
- Fair quality–compute comparison (Tab. R1),
- LatSearch outperforms EvoSearch under equal compute while being 4.8 times faster,
- Additional evaluation on Wan2.1-14B shows consistent gains (+1.03) (Tab. 3).

**Conclusion:** Fairness ensured; generalisation confirmed.

---

We sincerely appreciate the reviewers’ constructive feedback and have incorporated all their suggestions into substantial revisions, additional experiments, and clearer explanations. We hope this summary helps convey that all technical concerns were carefully addressed and that the revised submission is significantly strengthened.

Thank you again for your time and consideration.

Best regards,
Authors

---

### Meta-Review · Area_Chair_iEbb · 2026-01-08

**Summary:**

The submission presents an inference-time scaling framework for video generation that achieves higher video quality and greater efficiency, but the reviewers expressed concerns regarding the effectiveness of the Latent Reward Model and the sufficiency of the evaluation.

**Reviewer Concerns:**

Despite the authors’ clarifications and the additional experiments addressing technical details and metric robustness, critical concerns regarding the use of Latent MSE instead of Decoded MSE remain inadequately addressed. The higher error observed in Decoded MSE compared to Latent MSE indicates that there is an inherent gap between the intermediate latent and the final latent. This gap is only numerically reduced in latent space, but it indeed still exists.

**Reviewer Scores:**

The reviewers are likely to maintain their scores.

---

### Decision · Program_Chairs · 2026-01-26

Reject